

# Evolution of fault reactivation potential in deep geothermal systems. Insights from the greater Ruhr region, Germany

Michal Kruszewski[1], Alessandro Verdecchia[2], Oliver Heidbach[3,4], Rebecca M. Harrington[2], and David Healy[5]

[1]Chair of Engineering Geology and Hydrogeology, RWTH Aachen University, Lochnerstraße 4-20, 52056 Aachen, Germany
[2]Institute of Geology, Mineralogy, and Geophysics, Ruhr-University Bochum, Universitätsstraße 150, 44801 Bochum, Germany
[3]GFZ German Research Centre for Geosciences, Telegrafenberg, 14473 Potsdam, Germany
[4]Institute for Applied Geosciences, Technical University Berlin, Ernst-Reuter Platz 1, 10587 Berlin, Germany
[5]Department of Geology & Geophysics, School of Geosciences, University of Aberdeen, Aberdeen AB24 3UE, United Kingdom

**Correspondence:** Michal Kruszewski (kruszewski@lih.rwth-aachen.de)

**Abstract.** The success of deep geothermal systems depends on the presence of fault zones in the subsurface. Faults play a vital role in the Earth's plumbing system by facilitating fluid flow when they dilate, but are simultaneously known to enhance the hazard of the system once slipping in shear mode. As dilation of a fault enhances its permeability significantly, shear failure can lead to loss of boreholes or seismic events of economic concern. In this study, we present the evolution of reactivation

potential of major faults during 25-year production period in deep generic geothermal systems in the greater Ruhr region in western Germany. To determine the pre-operational *in situ* stress state we use a recently published comprehensive dataset of stress magnitude data from the greater Ruhr region in an analytical-probabilistic model accounting for uncertainties of *in situ* stress, fault geometry, and frictional properties for a prospective reservoir in the Devonian *Massenkalk* formations. The resulting cumulative distribution functions of dilation and slip tendency of given fault sets suggests that more than half of

the combined length of NW-SE-striking faults have a high reactivation probability, whereas the NE-SW-striking faults remain not optimally-oriented in the regional stress field. Using the relationship between dilation and slip tendency, we propose fault segments suitable for geothermal development that exhibit high hydraulic conductivity, i.e. high dilation tendency, and lower potential for shear failure, i.e. low slip tendency. In the second step, we employ generic thermo-hydro-mechanical models to quantify induced spatio-temporal stress changes on selected fault planes due to long-term geothermal production. We find

that after 25 years thermal stress changes contribute significantly to the change of the reactivation potential which should be accounted for while planning deep geothermal systems.

## 1 Introduction

The Earth's crust is characterised by a complex web of faults, fractures, and zones of inherent weakness, shaped by millions of years of tectonic activity. Although all faults in the subsurface have the potential to be reactivated, their distance to failure is

not uniform. The likelihood of fault reactivation depends on a number of factors, including the *in situ* stress conditions, fault





geometry and frictional properties (Morris et al., 1996; Zang and Stephansson, 2010; Blöcher et al., 2018). Anthropogenic activity, such as fluid injection or extraction during geothermal fluid production, can alter *in situ* stress conditions and induce fault reactivation, leading to either seismic or aseismic slip (Kim et al., 2018; Lei et al., 2019; Wang et al., 2020, 2021). On the other hand, faults provide underground fluid pathways and can increase the permeability of reservoirs, making them ideal

for deep geothermal energy exploitation, especially in reservoirs with low intrinsic matrix permeability (Caine et al., 1996; Younger et al., 2012).

Faults, although essential for creating an economically efficient geothermal system, may lead to unwanted failure by means of induced seismic events of economic concern (Grunthal, 2014) which can interrupt or even terminate geothermal fluid production (Häring et al., 2008; Edwards et al., 2015; Kim et al., 2018) or result in borehole failure due to shearing during drilling

operations (Zoback, 2010; Schmitt et al., 2012). For the above reasons, quantifying the complex interplay between faults, operational properties of a geothermal system, and the quantification of the spatio-temporal evolution of the distance to failure is crucial for the responsible and sustainable development of geothermal energy resources.

Fault stability studies are necessary to assess the likelihood of a given fault being reactivated (e.g., Walsh and Zoback, 2016; Seithel et al., 2019; Healy and Hicks, 2022). Although stability studies are useful, they highly depend on the amount and quality

of the input data. In cases of geothermal exploration campaigns with limited geological information, so-called geothermal greenfields, input data is burdened with high uncertainties, which can lead to poorly constrained estimate of the reactivation potential (i.e., the distance to failure). To provide a more realistic picture of the subsurface conditions, it is important to account for all known geological uncertainties, including structure, fault properties, and *in situ* stress conditions, utilizing probabilistic approaches.

In the greater Ruhr region in western Germany there is an abundance of high quality subsurface data due to more than 700 years lasting black coal mining activities. In 2018, the last mine ceased operation in the region, which initiated a discussion on how to repurpose the unused subsurface mining infrastructure for heat storage projects (Hahn et al., 2022). Furthermore, the utilization of deep geothermal resources located below the coal mining levels in the yet unexplored Lower Carboniferous and Devonian *Massenkalk* formations provide great potential to facilitate the green energy transition in this densely populated

region. The presence of Lower Carboniferous and Devonian carbonate rocks with karstification, fracture networks, and fault zones aligned with the regional stress orientation make the greater Ruhr region a promising candidate for large-scale deep geothermal energy use (Balcewicz et al., 2021; Kruszewski et al., 2021; Pederson et al., 2021; Lippert et al., 2022). Before 2018, the greater Ruhr region, although considered tectonically quiescent, experienced regular induced seismic events attributed nearly exclusively to the black coal extraction (Bischoff et al., 2010). Today, seismicity rates in the region are low and primarily

associated with either quarry blasting (RUB, 2007) or mine flooding activities (Rische et al., 2023). Seismic events caused by increasing water levels in abandoned coal mines associated with the latter commonly cause seismic events with local magnitudes, $M_L$, of up to 2.6 (Rische et al., 2023). Ultimately, the potential for geothermal energy exploitation presents an attractive opportunity for the greater Ruhr region, but simultaneously raises concerns regarding the fault reactivation potential and long-term stability associated with the implementation of deep subsurface systems.





This study presents a workflow to quantify the evolution of the reactivation potential associated with deep geothermal energy projects by means of dilation, $T_d$, and slip tendency, $T_s$. Since high $T_d$ values enhance the fluid flow and low $T_s$ values indicate a lower potential of shear failure, faults with high $T_d$ values and low $T_s$ are preferred settings. In a first step, we use an analytical-probabilistic model to quantify the contemporary distance to failure, or reactivation potential, accounting for ranges of known geological uncertainties due to the geometry, *in situ* stress conditions, and pore pressure, $P_p$. In a second step, we

build generic 3D thermo-hydro-mechanical (THM) models that quantify the induced spatio-temporal stress changes during 25 years of geothermal production on fault planes that represent the two prevailing fault orientations in the great Ruhr region.

    The results of this study provide critical information necessary for the safe and efficient development of deep geothermal projects and inform future planning efforts that need to balance the assessment of productivity and hazard assessment. Moreover, this work aims to identify hydrologically active fault systems, which could be potential targets for fluid extraction during

geothermal operations and have a relatively low probability for shear failure. Finally, our study will contribute to the acceleration of Germany's transition towards a green energy future, thereby helping to achieve the United Nations General Assembly's 2030 Agenda goals.

## 2   Geological setting

The greater Ruhr region is part of an external autochthonous fold and thrust belt of the late Variscan orogeny. Its main NW-

SE shortening direction developed throughout the Late Paleozoic convergence of Gondwanaland and Euramerican continental masses during Carboniferous and Devonian geological periods (Brix et al., 1988; Ziegler, 1990; Drozdzewski et al., 2009). The region was affected by tectonic activity in the Late Triassic, as well as by the Late Cretaceous transpression and Tertiary extension (Drozdzewski, 1993). Currently, the greater Ruhr region constitutes the western part of the Eurasian Plate with its present-day stress field resulting from a combination of a ridge push from the central and northern segments of the Mid-Atlantic

ridge and a northwards push of the African continent (Grünthal and Stromeyer, 1994).

    The greater Ruhr region is located between the Westphalian Lowlands in the north (part of the North German Plain; indicated in light green in Figure 1a), the Rhenish Massif in the south (marked with grey, brown, and dark green colours in Figure 1a), and the Lower Rhine Plain to the west (marked with yellow colour in Figure 1a). The region hosts two major fault network systems (Figure 1a). The first system is represented by NE-SW-oriented thrusts with steeply inclined to bed-parallel dip angles

reaching lengths of 40 km. Thrust faults have horizontal displacements of tens to hundreds of meters, with some reaching 2.5 km, and are dissected by NW-SE-striking post-Variscan normal faults resulting in a horst and graben structure (Gillhaus et al., 2003). Few strike-slip faults of various orientations are also present (Brix et al., 1988). Regional folds have variable dimensions and NE-SW-trending axes, wavelengths of up to 10 km that increase towards the north, and amplitudes of several hundred meters (Brix et al., 1988; Drozdzewski, 1988). As of today, the greater Ruhr region is under the influence of strike-slip

faulting regime (Kruszewski et al., 2022a). The regional maximum horizontal stress, $S_{Hmax}$, azimuth is 161 ±40°, with a slight clockwise rotation of $S_{Hmax}$ azimuth from west to east (Kruszewski et al., 2022a) (Figure 1).



Coal mining activities across the greater Ruhr region have exposed molasse-type clastic sediments of the Upper Carboniferous composed of shales, silt- to coarse-grained sandstones, and coal seams of variable strength, all heavily deformed by folding and thrusting (Bachmann et al., 1971). Cretaceous strata up to 1.8 km deep (Hesemann, 1965) overlay the Carbonif-

erous layers (Drozdzewski, 1993) north of the Ruhr region. Four deep exploratory boreholes (i.e., Münsterland 1 (Hesemann, 1965), Vingerhoets 93 (Eder et al., 1983), Versold 1, and Isselburg 3 (Drozdzewski, 1993)), located north of the Ruhr region all reach Devonian strata. The carbonate layers of the Middle and Upper Devonian period, which are part of the Devonian Reef Complex, outcrop south of the Ruhr region, close to the city of Iserlohn (Figure 1b). The results of the DEKORP 2N seismic line (DEKORP, 1990) indicated strong reflections that correspond to a high material contrast at around 5 km depth. The depth

of the material contrast is consistent with the depth of Devonian platform carbonates reached by deep boreholes north of the region (Drozdzewski, 1988; Franke et al., 1990; Drozdzewski, 1993). Due to the similarities with the Upper Jurassic carbonates of the Molasse basin in southern Germany, where geothermal operations are successfully ongoing (e.g., Przybycin et al., 2017; Moeck et al., 2019; Zosseder et al., 2022), the Early Carboniferous and Devonian carbonate formation located below the Ruhr region constitute potential deep geothermal reservoirs (Balcewicz et al., 2021; Kruszewski et al., 2021).

## 3  Methodology

We propose a two-stage approach for quantifying the fault reactivation potential (i.e., distance to failure) of deep geothermal systems. First, we develop an analytical-probabilistic model for the assessment of the contemporary fault stability accounting for all available geological data, including known model parameter uncertainties. Using the model, we calculate $T_s$, $T_d$, and fracture susceptibility, $S_f$, using conditional probabilities for each fault segment mapped in the region. As a next step, we

develop a coupled 3D THM model for two generic case scenarios representing two major fault orientations in the greater Ruhr region. For the first case, we simulate a geothermal doublet in the proximity of a fault with $T_s$ close to the reactivation threshold, whereas in the second case, we model a fault with $T_s$ significantly smaller than the reactivation threshold. We use the THM models to evaluate the spatio-temporal evolution of stress, pressure, and temperature during a planned life cycle of a deep geothermal system. Such an approach allows for a project-lifetime design based on a fault probability to slip. The sections

below detail both models.

### 3.1  Modelling the contemporary distance to failure with an analytical-probabilistic approach

We construct the analytical-probabilistic model using the updated version of the methodology developed in Healy and Hicks (2022), based on a combined Monte Carlo, response surface methodology, and Mohr-Coulomb theory, where $T_s$, $T_d$, and $S_f$ are computed for multiple fault segments simultaneously. $T_s$ defined as a ratio between the shear stress, $\tau$, and the effective

normal stress, $\sigma_n$', is given by the following (Morris et al., 1996)

$$T_s = \frac{\tau}{\sigma_n - P_p} = \frac{\tau}{\sigma_n'} , \qquad (1)$$



where, $\sigma_{\mathrm{n}}$ is the normal stress exerted on a fault plane and $P_{\mathrm{p}}$ is the pore pressure. $T_{\mathrm{d}}$ is computed by (Ferrill et al., 1999):

$$T_{\mathrm{d}} = \frac{\sigma_1' - \sigma_{\mathrm{n}}'}{\sigma_1' - \sigma_3'} \, , \tag{2}$$

where, $\sigma_1'$ and $\sigma_3'$ are the maximum and minimum effective principal stresses, respectively. Finally, $S_{\mathrm{f}}$, also known as

critical pore pressure, $\Delta P_{\mathrm{p}}$, being described as a change in pore fluid pressure needed to push given fault to failure, is defined as (Streit and Hillis, 2004)

$$S_{\mathrm{f}} = \Delta P_{\mathrm{p}} = \sigma_{\mathrm{n}}' - \frac{\tau - C_0}{\mu} \, , \tag{3}$$

where, $C_0$ is the cohesive strength of a fault and $\mu$ is the static friction coefficient. The computed values of $T_{\mathrm{s}}$, $T_{\mathrm{d}}$, and $S_{\mathrm{f}}$ of mapped fault segments derive from the distributions of several input parameters, including the magnitudes of vertical,

$S_{\mathrm{v}}$, minimum horizontal, $S_{\mathrm{hmin}}$, and maximum horizontal, $S_{\mathrm{Hmax}}$, stresses, the azimuth of $S_{\mathrm{Hmax}}$, and $P_{\mathrm{p}}$. Two additional input parameters were used for the case of $S_{\mathrm{f}}$ including $\mu$ and $C_0$. The input data used for the analytical-probabilistic model were extrapolated at the assumed reservoir depth based on the available geological data from the study region sourced from the Carboniferous layers (Table 1). We compute $T_{\mathrm{s}}$, $T_{\mathrm{d}}$, and $S_{\mathrm{f}}$ for a reservoir depth of $4.5\,\mathrm{km}$, i.e., depth of Devonian carbonates (DEKORP, 1990), separately for a set of NW-SE- and NE-SW-striking fault segments. We performed 10,000 simulations for

each fault segment, of the three scalar values and plotted the resultant cumulative distribution functions, *CDF*.

Each computed scalar value, $T_{\mathrm{s}}$, $T_{\mathrm{d}}$, and $S_{\mathrm{f}}$, has a corresponding threshold, i.e., a critical value. In the case of $T_{\mathrm{s}}$, we define a threshold value of 0.6 (Byerlee, 1978), which assumes fault reactivation for values of $T_{\mathrm{s}}$ that are greater or equal to the threshold. For $T_{\mathrm{d}}$, we assume a threshold value of 0.8, based on a case study by Ferrill et al. (2020a), which shows that mapped fractures with $T_{\mathrm{d}}$ exceeding 0.8 exhibit mineral coating, being indicative of crystal growth in an open void experiencing

dilation. The open void scenario is in contrast with slickenlined surfaces from the same study with $T_{\mathrm{d}}$ below 0.8 with negligible observed dilation. We assume a threshold of $0\,\mathrm{MPa}$ for the case of $S_{\mathrm{f}}$ which suggests that faults experiencing negative $S_{\mathrm{f}}$ values are unstable and subject to reactivation when subjected to insignificantly small stress changes. Stable faults, on the other hand, experience positive $S_{\mathrm{f}}$ values.

We extracted fault strike values from an open access map of major faults in the greater Ruhr region (GD NRW, 2017, 2019).

The data set include a total of 2347 fault segments, with a combined fault length of $2596\,\mathrm{km}$. For each fault segment, we selected a random dip angle based on a uniform distribution for both NW-SE- and NE-SW-striking faults with an assumption of a dip angle being between $65°$ and $85°$ for the former and between $35°$ and $60°$ for the latter. We used such distribution to account for the high uncertainty of fault geometry at the reservoir depth, and to represent all dip angles known from the available geological maps in the region (Jansen et al., 1986; Drozdzewski et al., 2007).

We used an *in situ* stress database from the greater Ruhr region based on an high number of 429 hydrofracturing tests to constrain distributions of $S_{\mathrm{hmin}}$, $S_{\mathrm{Hmax}}$, $S_{\mathrm{v}}$, and $S_{\mathrm{Hmax}}$ azimuth at the reservoir depth of $4.5\,\mathrm{km}$. Using the aforementioned data, we applied a skewed data distribution for both $S_{\mathrm{hmin}}$ and $S_{\mathrm{Hmax}}$ magnitudes, Gaussian distribution for $S_{\mathrm{v}}$ magnitude, and





Von Misses (circular) distribution for $S_{\mathrm{Hmax}}$ azimuth. We assume $P_{\mathrm{p}}$ following a Gaussian distribution and being equal to the hydrostatic pressure of a cold water column with density of $1000\,\mathrm{kg\,m^{-3}}$. For the computation of $S_{\mathrm{f}}$, both $\mu$ and $C_0$ were

assumed to follow Gaussian distributions. Table 1 shows the input parameters used for establishing data distributions, and Figure 2 shows an example of data distributions used for $T_{\mathrm{s}}$ computation of the NW-SE-striking fault segments.

### 3.2   Modelling the evolution of the distance to failure with THM model

We define two generic settings for the development of a deep fault-based geothermal system and simulate THM processes during 25 years of geothermal production and 25 years of reservoir reaching equilibrium state without fluid circulation in a

doublet system. The selection of the two case scenarios was based on the two extreme cases of fault being close to critical state and a fault being far away from failure, which also mirrors the two prevailing fault orientations in the greater Ruhr region. Such analysis constitutes an investigation of spatio-temporal changes of $P_{\mathrm{p}}$ and stress during long-term injection/production operations. We model the single-phase fluid flow in faults following Darcy's law, which are represented as 2D planes of weakness cutting the 3D poro-elastic model volume. Fluid is injected into the reservoir through an injection well, and produced from a

production well within a so-called geothermal doublet under the mass balance principle. We perform the numerical simulation and model discretization with the *COMSOL Multiphysics* software (COMSOL, 2021), which accounts for $P_{\mathrm{p}}$ and associated stress and strain changes. The applied fluid-flow governing equations result from the conservation of momentum and mass in a fully saturated porous medium, whereas the heat transport governing equations are derived from the heat balance that results from both advective and conductive heat transport. We refer the reader to further details on the theory behind how fluid circu-

lation affects effective stresses on faults within a framework of coupled THM simulations in COMSOL (2021). Effects such as fault permeability enhancement due to the dilation, change of rock properties due to $P_{\mathrm{p}}$ or temperature, $T$, the influence of fluid chemistry on rock mass and fault properties, mechanisms of earthquake interactions, and the Kaiser effect are not considered in the simulation.

We assume fault reactivation based on the Coulomb friction law and the notion that shear slip is controlled by the ratio

of shear stress, $\tau$, to effective normal stresses, $\sigma_{\mathrm{n}}{}'$, acting on a fault plane in the prevailing stress field, and coupled with thermo-poro-elastic response of the porous medium to the fluid circulation. The assumptions imply that fault reactivation will be initiated once $T_{\mathrm{s}}$ exceeds the frictional resistance, $\mu$, of a fault. The size of the reactivated fault area will be restricted to the size of the critically stressed fault patch.

Within each case scenario, we selected two locations for our case scenarios, including i) a fault with strike of $140°$, north-

eastward dip angle of $60°$, and an initial $T_{\mathrm{s}}$ of 0.57 (referred to as 'NW-SE-striking fault'), and ii) a fault with strike of $50°$, southeastward dip angle of $40°$, and an initial $T_{\mathrm{s}}$ of 0.18 (referred to as 'NE-SW-striking fault'). The simplified reservoir model has a dimensions of $20\,\mathrm{km}$ by $20\,\mathrm{km}$ by $8.7\,\mathrm{km}$ and includes three stratigraphic units: impermeable overburden, representing Carboniferous layers; reservoir, representing carbonate *Massenkalk* formations of improved matrix permeability; and an impermeable underburden representing mid to early Devonian layers (Figure 3a). A fault bisects the centre of the model volume

between two vertical open-hole sections of production and injection wells, straddling the fault at similar distance. The wells





are separated by $1\,\mathrm{km}$ and have a final depth of $4.3\,\mathrm{km}$ and a length of an open-hole section of $0.2\,\mathrm{km}$ (Figure 3b). The fault has a length of $7\,\mathrm{km}$ and width of $9\,\mathrm{km}$ and is assumed to extend from approximately $0.2\,\mathrm{km}$ below the surface.

We explored two different models to assess the significance of thermally induced stresses during long-term geothermal production. The first model does not account for the thermal influence resulting from an injection of a cold fluid into a hotter
reservoir (isothermal model). The second model accounts for the thermal influence, where fluid flows in the subsurface with non-constant temperatures (nonisothermal model).

We assume $\mu$ of 0.6 and a cohesionless fault (Byerlee, 1978) and a geothermal gradient of $35\,^{\circ}\mathrm{C}\,\mathrm{km}^{-1}$ (Wedewardt, 1995), with the surface temperature of $10\,^{\circ}\mathrm{C}$ and injection temperature of $50\,^{\circ}\mathrm{C}$. We model fluid circulation within a geothermal system starting in the first year of operation with duration of 25 years. We model the spatio-temporal evolution of stress for a
duration of 50 years, with a time between production cessation and 50 years being considered as a period in which the reservoir returns to equilibrium conditions. Injection fluid is assumed to be clean water with its properties being temperature-dependent. Table 2 lists all other material properties assigned to the specific geological layers and the fault.

We initiate the *in situ* stress state within the model using the relations between $S_{\mathrm{v}}$ and $S_{\mathrm{hmin}}$, as well as between $S_{\mathrm{v}}$ and $S_{\mathrm{Hmax}}$ based on the mode values from Table 1. The magnitudes of the initial stress tensor are, therefore, represented by linear
gradients and are artificially distributed across the model volume. The initial stress tensor constitute, therefore, merely a model input parameter. We assume the $S_{\mathrm{Hmax}}$ azimuth in the numerical model to be equal to the circular mean value from Table 1. The bottom and side model boundaries are free to move in-plane and are fixed in the out-of-plane direction, whereas the surface is free of constraints. We impose a hydraulic head on all sides of the model and a fixed atmospheric pressure of $0.1\,\mathrm{MPa}$ at the surface. The open hole sections of both wells are 1D line sources, where fluid injection and production are defined with
mass-flow rates and where the injection well serves as a line heat source. The model sides are open boundaries for heat transfer, whereas we fix a constant temperature at the surface and bottom boundaries, with values of $10\,^{\circ}\mathrm{C}$ and $316\,^{\circ}\mathrm{C}$ respectively.

## 4   Results

### 4.1   Contemporary distance to failure

This section presents results of $T_{\mathrm{s}}$, $T_{\mathrm{d}}$, $S_{\mathrm{f}}$, as well as the relationship between $T_{\mathrm{s}}$ and $T_{\mathrm{d}}$ and respective failure modes from the
analytical-probabilistic model.

#### 4.1.1   Slip tendency

Figure 4a and Figure 4b show the results of $T_{\mathrm{s}}$ calculations as *CDF* plots. Each line in Figure 4a and Figure 4b represents one fault segment of a certain length corresponding to a fault segment presented on the map view (Figure 4c). In the *CDF* plots, more stable fault segments, i.e., ones with lower $T_{\mathrm{s}}$, skew towards the left, whereas less stable fault segments, i.e., ones with
higher $T_{\mathrm{s}}$, skew to the right. The pink shaded areas in Figure 4a and Figure 4b show the expected range of $\mu$, being between 0.6 and 0.85 (Byerlee, 1978). The red fault segments have a conditional probability of at least 33% of their $T_{\mathrm{s}}$ above the threshold





value of 0.6 (where $\mu$ is marked with a thick vertical black line in Figure 4a and Figure 4b), the amber fault segments have probability between 1% and 33%, and blue fault segment have less than 1% probability of being unstable.

As fault segments considered in the analysis have variable length, we refer to their combined length for further analysis. From the total combined fault length, 53% have 33% or higher probability of exceeding threshold friction, 34% have a between >1% and <33% probability, whereas 13% have a less than 1% probability. For the case of the combined length of the NW-SE-striking faults exclusively, 57% have >33% probability of exceeding threshold friction, 36% have between >1% and <33% probability, whereas 7% have <1% probability of being unstable. In the case of the combined length of NE-SW-striking faults, none have >33% probability of exceeding threshold friction, only 1% have between >1% and <33% probability, and 99% have <1% probability of being unstable.

### 4.1.2   Dilation tendency

Figures 5a and 5b show the $T_d$ calculation results as *CDF* plots and Figure 5c in map view. The *CDF* plots show fault segments with lower $T_d$ that are less prone to dilate and transmit fluids, skewed towards the left, whereas fault segments that are more likely to dilate and assist fluid flow with higher $T_d$, skewed to the right. From the total combined fault length, 63% have a probability of 33% or higher of exceeding the $T_d$ threshold of 0.8 (marked with a thick vertical black line in Figures 5a and 5b), 5% have a a probability between >1% and <33%, and 32% have a probability of less than 1%. For the case of the combined length of only NW-SE-striking faults, 67% have a probability of >33% of exceeding $T_d$ of 0.8, 5% have probability between >1% and <33%, and 28% have a probability <1%. In the case of the combined length of NE-SW-striking faults, 100% have a probability <1% of exceeding $T_d$ of 0.8.

### 4.1.3   Fracture susceptibility

Appendix A1a and A1b show the results from the $S_f$ calculations as *CDF* plots and Appendix A1c show the results in map view. In the *CDF* plots, reactivation-prone fault segments that need negligible additional pressure to become active, or have negative $S_f$ or values equal to zero, skew towards the left. Stable fault segments requiring high pressures for reactivation with positive $S_f$ values skew to the right.

From the total combined fault length, 29% have a probability of 33% or higher of $S_f$ being negative or equal to zero, 53% have a probability between >1% and <33%, and 18% have a probability of less than 1%. For the case of the combined length of the NW-SE-striking faults exclusively, 31% have probability >33% of $S_f$ being negative or equal to zero, 57% have probability between >1% and <33%, and 12% have probability <1%. In the case of the combined length of NE-SW-striking faults, no fault has probability >33% of $S_f$ being negative or equal to zero, 3% have probability between >1% and <33%, whereas 97% have <1% probability.



### 4.1.4  Failure modes

Several authors have shown that the relationship between $T_s$ and $T_d$ can be considered an efficient indicator of fault deformation behavior and could suggest either conduit or sealing behavior of faults (Ferrill et al., 2020b; Bhowmick and Mondal, 2022).
Figure 6 demonstrates the relationship between $T_s$, $T_d$, and associated rock failure modes and volume change for all fault
segments considered in this study. For computation of $T_s$ and $T_d$ in Figure 6, we have used a deterministic approach and assumed mode and mean values from the second column of Table 1 as input parameters. We divide Figure 6a and Figure 6b into five failure (or reactivation) mode regions (tensile, hybrid, shear, compactive shear, and compactive). We apply arbitrary values of $T_d$ of 0.4, 0.7, and 0.9 to discriminate between different failure mode regions. In addition, we discriminate between regions where faults experience volume gain, loss, or are volume neutral. Cross-plotting the calculated $T_s$ and $T_d$ of all fault
segments shows a broad spectrum of both $T_s$ and $T_d$ values. Most of the NW-SE-striking fault segments fall into hybrid as well as tensile failure modes that experience volume gain and exemplify high $T_d$ and $T_s$ values. Conversely, the NE-SW-striking fault segments exhibit exclusively compactive failure and volume loss, exemplifying low $T_d$ and low $T_s$ values.

### 4.2  Evolution of the distance to failure

Figure 7a and Figure 7b, respectively show the perturbed temperature and pressure fields during geothermal production along
the shortest distance between injection and production wells for the case of a nonisothermal (i.e., $\Delta T \neq 0$) THM model with a NW-SE-striking fault with initially high $T_s$. Changes of $P_p$ and temperature are both diffusive processes, but they act on very different spatio-temporal scales. The development of the thermal front in the reservoir is a relatively slow process, which does not cease once fluid circulation in a geothermal system has ended. The thermal front for the case presented in Figure 7a does not exceed 500 m. The pressure front migration, in contrast, is comparatively faster, where the reservoir achieves equilibrium
pressure conditions in a relatively short period after circulation initiation. Moreover, the pressure front abates relatively quickly after circulation between the wells is stopped (evidenced by a constant pressure between injector and producer at year 50). The pressure gradient between injection and production well (Figure 7b) becomes more asymmetric with time (i.e., higher pressure at the injection well at the end of fluid circulation due to higher viscosity of the colder injection fluid) with the neutral point being relatively stable throughout the simulation time. The neutral point is located around the mid-point between wells, which
coincides with the fault location (at distance of 500 m from the injection well).

Figure 7c presents the cumulative reactivated area ($A_r$) (i.e., fault area where $T_s \geq 0.6$) on the NW-SE-striking fault, for the case of both nonisothermal (i.e., $\Delta T \neq 0$) and isothermal (i.e., $\Delta T = 0$) numerical models. Strikingly, the isothermal model shows no fault reactivation throughout the simulated period, which means that none of the fault area reached the failure criterion. Fault reactivation starts in the nonisothermal model within the first few years after circulation initiation, and increases
quasi-linearly until circulation ceases in the 26[th] year. Interestingly, the maximum $A_r$ is not achieved during or in the last year of fluid production (i.e., at 26[th] year), but rather, after circulation cessation, at the 30[th] year, and amounts to 5.2E+5 m². After the peak, $A_r$ decreases slightly, reaching 5E+5 m² at the end of simulation. The change in $A_r$ implies that the thermal stress



contribution is vital for the long-term stability stress changes of geothermal systems and may contribute significantly to the distance of failure of major regional faults.

Figure 8 shows the spatio-temporal changes of shear stress, $\Delta\tau$, normal effective stress, $\Delta\sigma'_n$, and $\Delta T_s$ in years 5, 25, and 50 for the case of the nonisothermal numerical model with the NW-SE-striking fault. The injected water cools the rock mass in the vicinity of the injection well, inducing contraction of the rock mass. The contraction results in negative volumetric strain, which then drives the change of $\tau$ and $\sigma'_n$ on the fault plane. The area cooled by injection expands with time during geothermal production as well as after the production is ceased. The areas of stabilization (where $\tau$ and $T_s$ decrease and $\sigma'_n$ increases),

represented with blue colours in Figure 8, and destabilization areas (where $\tau$ and $T_s$ increase and $\sigma'_n$ decreases), represented with red colours in Figure 8, correlate with the destabilizing effects of an injection well and stabilizing effects of production well, respectively. Both are observable on the fault plane. The destabilised area on the fault increases significantly between year 5, where it is mainly localised within the reservoir, and year 25, where it extends beyond the reservoir and into the under- and overburden. There is no significant variation between year 25 (end of circulation), year 50 (end of the simulation), in either

$\Delta\tau$, $\Delta\sigma'_n$, or $\Delta T_s$.

Finally, for the case of the NE-SW-striking fault with initially low $T_s$, we observed no reactivation for either nonisothermal ($\Delta T \neq 0$) or isothermal ($\Delta T \neq 0$) numerical models (Appendix A2). The lack of reactivation means that the perturbed *in situ* conditions during fluid circulation between the injection and production wells were not high enough for any of the fault segments to meet or exceed the assumed frictional threshold of 0.6.

## 290    5    Discussion

In this study, we developed a workflow to quantify the evolution of the distance to failure of major faults associated with geothermal production, including known geological uncertainties prior to any subsurface operations. Our methodology combines probabilistic and numerical modelling techniques to discriminate areas of high induced seismic hazard allowing geothermal systems to be selected within regions of higher fault stability. The approach offers promise for achieving wider public

acceptance of subsurface geothermal systems by mitigating the possibility of felt and damaging induced seismic events. With this study, we show that several faults have a higher probability to be intrinsically critically-stressed (red-coloured fault lines in Figure 4), where small stress perturbations could lead to fault reactivation. However, fault orientations with reasonable operational windows exist, and may enable establishing stable geothermal systems with lower induced seismic hazard potential (amber and green-coloured fault lines in Figure 4). Such fault segments could be considered as potential targets for deep

geothermal systems in the region.

### 5.1    Analytical-probabilistic approach

The results of the analytical-probabilistic model show that a large number of NW-SE-striking faults at the reservoir depth in the greater Ruhr region have high reactivation probability (Figure 4a, c) and exhibit high dilation (Figure 5a, c). Previous studies on mining-induced seismicity and fault permeability explored during coal mining operations are consistent with such observations



(Thielemann and Littke, 1999; Reinewardt et al., 2009; Peña-Castro et al., 2018). More recently, a series of earthquakes were recorded during the flooding of an abandoned coal mine in the eastern Ruhr region, close to the city of Hamm (Rische et al., 2023). The largest observed earthquake (2.6 $M_{\mathrm{L}}$) and occurred during the rise of the mine water table level of approximately 100 m, which is equivalent to a hydrostatic column pressure change of around 1 MPa. Small fluid pressure changes on the order of 1 MPa in the previously drained rock mass were sufficient to induce seismic events of felt magnitudes and are potentially

related to the critically-stressed faults. Conversely, the NE-SW-striking faults exhibit low reactivation probabilities (Figure 4b, c) and low dilation (Figure 5b, c).

The joint analysis of $T_{\mathrm{s}}$ and $T_{\mathrm{d}}$ represents a unique opportunity to evaluate the potential of existing faults for circulating fluids and their distance to failure. The majority of the NW-SE faults in the region are expected to be in a either hybrid or tensile failure regime with $T_{\mathrm{d}}$ exceeding the threshold value of 0.8, which could suggest that in the reservoir, faults and fractures are

hydraulically open and can be (re-)mineralised. Therefore, chemical stimulation could be more efficient for inducing fluid flow within NW-SE faults and fractures in comparison to e.g., hydraulic stimulation. The NW-SE-striking faults are more prone to exceed critical $T_{\mathrm{s}}$ values, which results in a higher intrinsic seismic hazard potential. In contrast, the NE-SW-striking faults exhibit compacting failure modes, and are potentially hydraulically closed. Structures with low $T_{\mathrm{s}}$ and $T_{\mathrm{d}}$ values could induce reservoir compartmentalization and act as barriers to fluid flow, while having at the same time low probability of reactivation.

**5.1.1 Reduced-risk dilation tendency**

To ease the quantification of faults that should be favoured for geothermal energy extraction based on combined analysis of their $T_{\mathrm{s}}$ and $T_{\mathrm{d}}$, we define an additional scalar value called reduced-risk dilation tendency

$$T_{\mathrm{dn}} = T_{\mathrm{d}} \left( 1 - \frac{T_{\mathrm{s}}}{max(T_{\mathrm{s}})} \right) , \tag{4}$$

where, $max(T_{\mathrm{s}})$ is the maximum $T_{\mathrm{s}}$ value in a given *in situ* stress conditions. The $T_{\mathrm{dn}}$ value ranges between 0 and 1, where

0 indicates fault with lowest $T_{\mathrm{d}}$ and highest $T_{\mathrm{s}}$, and 1 indicates a fault with highest $T_{\mathrm{d}}$ and lowest $T_{\mathrm{s}}$. Fault segments with high $T_{\mathrm{dn}}$ are more prone to reactivation in opening mode rather than shear mode, which will potentially result in a lower seismic hazard and potentially higher permeability. Such conditions are ideal for geothermal exploration. Faults with low $T_{\mathrm{dn}}$ have a higher probability to be reactivated in shear mode (seismic or aseismic), and therefore should be avoided during geothermal exploration. Figure 9 presents the computed $T_{\mathrm{dn}}$ for all faults in the greater Ruhr region using a deterministic approach and

assuming the same input parameters used to compute $T_{\mathrm{s}}$ and $T_{\mathrm{d}}$ from Figure 6. The figure demonstrates that very few fault segments have favourable stress conditions for development of a geothermal system, where favourable faults strike parallel to the azimuth of $S_{\mathrm{Hmax}}$ and have steep dip angles. $T_{\mathrm{dn}}$ could therefore be a useful tool for assessing how favourable a given fault could be for a development of a geothermal system.





### 5.1.2 Limitations and uncertainties

Although our study presents a promising picture of fault stability at the prospective reservoir depths in the greater Ruhr region, there are still significant limitations to the analysis. The analytical-probabilistic model results are strictly dependent on the availability, amount, and quality of the input data. The fault dataset extracted from available geological maps in the region (GD NRW, 2017, 2019) represents fault geometries sourced from the Carboniferous units only. There is a lack of geological information from the Devonian layers, with the exception of a few deeper wells in the Münsterland region, north of the greater

Ruhr region (e.g., Hesemann, 1965). The fault geometries analysed in this study could vary relative to the ones at reservoir depths (i.e., different lengths, strike, and dip angles) or be even absent altogether. However, recent geological work from the Ruhr region (Balcewicz et al., 2021; Pederson et al., 2021) shows the strong presence of NNW-SSE, and NW-SE oriented faults and fractures in the Devonian *Massenkalk* outcrops, suggesting that the faults analysed in this work could theoretically extend to depths beyond the Carboniferous layers. The presence of additional unmapped faults at reservoir depths should not

be excluded. Further 2D and 3D seismic campaigns, as well as deep exploratory drilling ventures would aid in constraining the fault architecture and reservoir geometry at depth.

The assumption of $\mu$ and cohesionless faults made in this study are based on common assumptions from the literature (i.e., Byerlee, 1978; Zoback, 2010), but could differ at reservoir depths. Different types of fault core material could either decrease (e.g., clay) or increase (e.g., calcite) $\mu$ and fault cohesive strength. Unfortunately, the lack of available information

on the frictional properties of carbonate rocks in the region makes it significantly difficult to constrain frictional properties. Laboratory studies in reservoir-like conditions are needed to better constrain the frictional properties in the greater Ruhr region. The assumption of $T_d$ threshold value of 0.8 (Ferrill et al., 2020b), indicative of a re-mineralised fault, should be confirmed by local outcrop studies correlating the movement of fractures and faults (e.g., slickensides) and the filling material (e.g., veins) together with the historical state of stress.

The *in situ* stress magnitude and fault orientation database available in the greater Ruhr region (Kruszewski et al., 2022b) used to constrain inputs values for the analytical-probabilistic models present a comparatively unique and high density data set relative to other regions where it is known that only negligible information on the stress state is available (Heidbach et al., 2018). Despite the high density of stress data and efforts to filter out stress magnitudes lower than the assumed hydrostatic $P_p$ conditions, effects of water drainage, due to the pumping water out of the coal mines, as well as compaction effects, could

affect stress magnitudes measured during hydrofracturing in deep mines (Niederhuber et al., 2022; Kruszewski et al., 2022b). If so, certain stress measurements used in our study to constrain model inputs, could be characterised by a perturbed stress status field rather than a virgin one.

Similar to fault geometry, data of stress orientation and magnitude come exclusively from Carboniferous layers due to a lack of data coming from Devonian layers. Although we do not expect significant changes in the stress gradients at the reservoir

depth in comparison to the overburden (Zoback, 2010), the possibility of non-linear changes in stress magnitudes should not be excluded, even due to the higher stiffness of the reservoir rocks in comparison to the over- and underburden layers (Table 2). Hydraulic tests in deep exploratory boreholes are urgently needed to constrain $P_p$, whereas 'direct' (i.e., field) measurements





such as e.g., hydrofracturing, overcoring tests or borehole imaging logs are necessary to constrain principal stress magnitudes, and principal stress orientations at the reservoir depths (Ziegler and Heidbach, 2022).

## 5.2 THM model

The THM models presented in this study show that the contribution of $P_\mathrm{p}$ changes to fault instability is significant, especially at the beginning of the geothermal production and in a case of a system being close to a fault segment. However, the contribution of thermal stress due to the thermal contraction of the rock mass is the dominant control on the size of the reactivated fault area during the long-term geothermal operations. Fault reactivation can therefore not be explained exclusively with pressure diffusion and related $P_\mathrm{p}$ changes. The thermal effects resulting from long-term subsurface operations must be taken into account. With the coupled numerical models (Figure 7 and Figure 8), we show that it is not only the changes of $\sigma'_n$, but also $\tau$ changes that are affected during geothermal production. Therefore, accounting for fluid pressure contribution and change of $\sigma'_n$ only could lead to an over-simplified picture of fault stability. Scalar values used for fault stability evaluation based on the contribution of fluid pressure only, such as $S_\mathrm{f}$, will not provide a full picture of the fault stability *in situ*. As the evolution of the thermal front progresses slowly compared to the pore pressure front, the rate of induced seismic events will depend primarily on the size of the thermal front and its distance to the closest faults. The methodology presented in this study, could allow the planning of a safer and more stable location for a deep geothermal system and potentially increase its lifetime based on an induced seismic hazard criteria. It therefore presents a powerful tool during especially exploration and appraisal project phases of the geothermal system development.

As evidenced by the numerical models, NW-SE-striking faults are much more prone to reactivation during long-term fluid circulation within a geothermal doublet. In contrast, the NE-SW-striking faults stay relatively stable with stress conditions below the frictional thresholds throughout the simulation period. Methods for the reduction of localised stress and pressure concentrations for the future design of geothermal systems based on the likely-to-be-active NW-SE-striking fault segments might include use of multilateral borehole design or horizontal drilling to increase the size of contact between the open hole section of the injection well and the reservoir.

### 5.2.1 Limitations and uncertainties

The numerical models presented here provide good first-order estimation of the spatio-temporal evolution of stress and seismicity and present a pre-operational picture of the fault reactivation potential of a deep geothermal system for a specific case of rock mass and operational parameters (Table 2) in the greater Ruhr region. Model results are still burdened by limitations and uncertainties. Models developed in this study need to be calibrated and validated based on new geological data, such as results from seismic campaigns and exploratory drilling ventures.

The geometry of geological layers, as well as fault architecture, was purposely oversimplified to achieve computational efficiency and provide a first-order estimation of the induced seismic hazard of generic deep geothermal systems. The actual geometry of a fault, with its specific offsets, thickness, permeable damage zone, and impermeable fault core, as well as folded geometry of different geological units, will affect the distribution of stresses exerted on a given fault plane during fluid



circulation. An assumption of an isolated fault, as exercised in our study, is a significant simplification. In reality, a web of multiple faults and fractures will be present, with some acting as fluid conduits and some as barriers to flow. Conduit-barrier behavior will affect the redistribution of fluid flow and, thus, stress conditions on faults. The above-mentioned effects should be, however, evaluated for a specific area of interest, once results from 2D or 3D seismic campaigns supported by results from exploratory drilling are available.

A major simplification of the numerical models developed in this study involves the assumption of a homogeneous isotropic permeability tensor within the reservoir and in over- and underburden. An assumption of homogeneous isotropic permeability tensor means that the pressure change in the reservoir spreads out uniformly from an injection well. In reality, systems of dense natural fracture networks (Balcewicz et al., 2021; Kruszewski et al., 2021) and karstifications (Pederson et al., 2021) expected at depths of the Devonian layers in the greater Ruhr region will result in an anisotropic permeability tensor. Anisotropic permeability tensor will influence the fluid flow in the reservoir. Geological complexities could result in large pressure differences at significant distance from an injection well compared to models with an isotropic permeability tensor. Since there are no published studies in the region in which systems of natural fractures or karsts were mapped or modelled, the homogeneous isotropic permeability tensor of the reservoir rock is still considered a good first-order approximation. As assumed values of reservoir and fault permeability (Table 2) remain relatively uncertain, it is advised to carry out pressure transient testing in deep exploratory boreholes to extract the permeability values and understand the pressure response of the reservoir.

The probability of fault reactivation is different than the probability of an induced seismic event of a certain magnitude as it depends on the lateral dimension of a fault and rupture dynamics where other stress-changing processes have to be taken into account. The predicted cumulative reactivated area, as presented in 7c, raises the probability of a seismic event nucleating on a particular fault, but it cannot be directly translated into a magnitude of a seismic event. One should also not exclude that the stress and strain accumulated by a fault could be released in a form of an aseismic slip.

## 6 Conclusions

In this study, we developed a workflow for the quantification of the pre-operational fault reactivation potential of deep geothermal systems accounting for the known ranges of geological uncertainties. Our proposed methodology, combining probabilistic and numerical modelling techniques, allows for deep geothermal systems to be selected within regions of higher permeability and lower seismic hazard. The developed workflow presents a promising solution for achieving wider public acceptance by lowering the possibility of large induced seismic events of economic concern and reducing occurrence probability of induced seismic events down to acceptable levels. As the reliability of fault stability studies highly depends on the amount and quality of the model input, we strongly recommend acquiring more data from the depths of interest (i.e., geothermal reservoir) including primarily the *in situ* stress state information, detailed fault architecture, and reservoir-specific rock properties.

Results of the probabilistic analysis prove that the NW-SE faults are much more susceptible to reactivation than the NE-SW thrusts in the greater Ruhr region, where the former are potential reservoir targets and the latter could cause compartmentalisation. The majority of NW-SE faults are expected to be reactivated in either hybrid or tensile mode, whereas the NE-SW faults





are almost exclusively in compacting failure modes. Based on the results from the numerical modelling, we show that although
the contribution of the pore pressure to the fault stability is significant, it is the thermal stress resulting from rock contraction
effects of the colder injection fluid in the hotter rock mass that primarily controls the size of the reactivated fault area during
long-term geothermal production.

*Code availability.*  The code used in this study is available at https://github.com/DaveHealy-github/pfsRuhrBochum, last access: 1st of August 2023.

*Data availability.*  The shapefiles of faults with computed slip tendency, dilation tendency, fracture susceptibility, and reduced-risk dilation
tendency based on this study, as well as the *COMSOL Multiphysics* thermo-hydro-mechanical models used to compute the spatio-temporal
evolution of stress conditions *in situ* are available in Kruszewski and Verdecchia (2023).

## Appendix A:  Slip, dilation tendencies and fracture susceptibility

### A1   Slip tendency

Reactivation of a fault depends on its geometry and acting stress state conditions (Barton et al., 1995). Once geometry and
stress state are known, shear, $\tau$, and effective normal, $\sigma_\mathrm{n}$', stresses exerted on any arbitrarily oriented fault can be resolved
(Jaeger et al., 2007). Subsequently, in accordance with the Amonton's law, slip tendency, $T_\mathrm{s}$, defined as a ratio between $\tau$ and
$\sigma_\mathrm{n}$', can be assessed (Morris et al., 1996)

$$T_\mathrm{s} = \frac{\tau}{\sigma_\mathrm{n} - P_\mathrm{p}} = \frac{\tau}{\sigma_\mathrm{n}'} \; . \tag{A1}$$

where, $\sigma_\mathrm{n}$ is the normal stress exerted on a fault plane and $P_\mathrm{p}$ is the pore pressure. Cohesionless (i.e., $C_0 = 0$) faults with $T_\mathrm{s}$
approaching or exceeding frictional coefficient of sliding friction, $\mu$, have increased likelihood for a shear movement and could
be deemed "unstable". The $T_\mathrm{s}$ embodies the principle underlying Mohr-Coulomb shear failure, i.e., increasing $\tau$ relative to $\sigma_\mathrm{n}$'
causes fault to approach and eventually slip. It is assumed that fault reactivation will be initiated once $T_\mathrm{s}$ will exceed $\mu$, which
in this study is assumed to be equal to 0.6 (Byerlee, 1978)

$$T_\mathrm{s} \geq 0.6 \; . \tag{A2}$$

### A2   Dilation tendency

To investigate the tendency of a fault to dilate (i.e., open) under the *in situ* stress state, dilation tendency, $T_\mathrm{d}$, is computed
(Ferrill et al., 1999)





$$T_\mathrm{d} = \frac{\sigma_1' - \sigma_\mathrm{n}'}{\sigma_1' - \sigma_3'} \,, \tag{A3}$$

where, $\sigma_1'$ and $\sigma_3'$ are the maximum and minimum effective principal stresses, respectively. It was assumed that in the case of $T_\mathrm{d}$ exceeding 0.8, given fault may exhibit crystal growth in an open void experiencing dilation (Ferrill et al., 2020a)

$$T_\mathrm{d} \geq 0.8 \,. \tag{A4}$$

## A3   Fracture susceptibility

Fracture susceptibility, $S_\mathrm{f}$, also known as critical pore pressure, $\Delta P_\mathrm{p}$, is the change in pore fluid pressure needed to 'push' a
given fault to failure and is defined as (Streit and Hillis, 2004)

$$S_\mathrm{f} = \Delta P_\mathrm{p} = \sigma_\mathrm{n}' - \frac{\tau - C_0}{\mu} \,. \tag{A5}$$

where, $C_0$ is the cohesive strength of a fault. $S_\mathrm{f}$ of $0\,\mathrm{MPa}$ or higher will indicate an unstable fault which is likely to be reactivated if experienced even small stress changes

$$S_\mathrm{f} \geq 0 \,. \tag{A6}$$

**A4   Failure modes**

The transition from tensile to hybrid, shear, compactive shear, and compactive fault failure mode corresponds to $T_\mathrm{d}$ transition from 1 to 0, where failure along fault surfaces normal to $\sigma_3$ having $T_\mathrm{d}$ of 1 (i.e., volume gain), and fault surfaces normal to $\sigma_1$ having $T_\mathrm{d}$ of 0 (i.e., volume loss). With respect to $T_\mathrm{s}$, shear failure corresponds to high $T_\mathrm{s}$ and moderate $T_\mathrm{d}$ values. Although such type of failure is often considered volume neutral, in reality shear failure has associated tensile or hybrid
failures, which provide conductive pathways for fluids (Barton et al., 1995). The transition towards hybrid or tensile failures is marked by reduction of $T_\mathrm{s}$ and increase in $T_\mathrm{d}$, culminating in $T_\mathrm{d}$ being equal to 1. On the other hand, the transition from shear to compactive shear and compactive failures is marked by reduced $T_\mathrm{s}$ and $T_\mathrm{d}$ culminating in both $T_\mathrm{s}$ and $T_\mathrm{d}$ values being equal to zero (Ferrill et al., 2020b, a).

*Author contributions.* MK - conceptualisation, methodology, validation, formal analysis, investigation, data curation, writing (original draft,
review, and editing), visualisation, supervision. AV - conceptualisation, methodology, validation, investigation, writing (original draft, review, and editing), visualisation. OH - writing (review and editing). RMH - writing (review and editing). DH - conceptualisation, methodology, validation, writing (review and editing).



*Competing interests.* At least one of the (co-)authors is a member of the editorial board of Solid Earth.

*Disclaimer.* The authors reserve the right not to be responsible for the topicality, correctness, completeness, and quality of the information
provided. Liability claims regarding damage caused by the use of any information provided will be rejected. Conclusions made in this study
are solely opinions of the authors and do not express the views of the employer, university, or funding agency.

*Acknowledgements.* Authors would like to thank Alireza Teimouri Jervekani for his help during the preparation of this manuscript. Special
thanks go to Giordano Montegrossi from CNR for intensive discussions and help with interpretation of numerical modelling results.



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



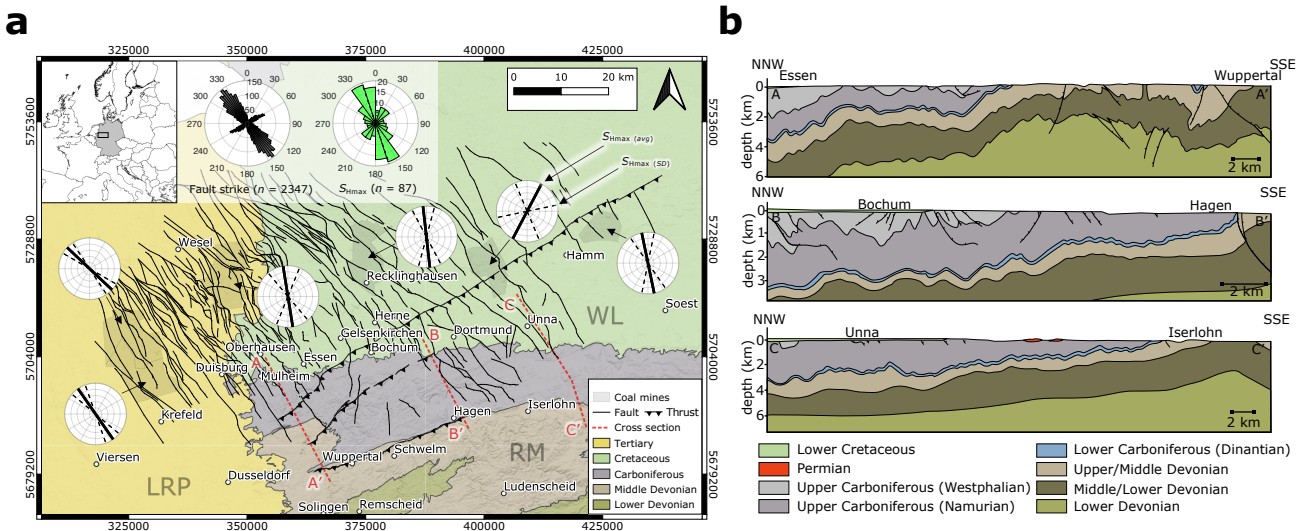

**Figure 1.** a) Map of the greater Ruhr region with major fault zones (after GD NRW 2017, 2019) with rose plots representing circular mean azimuths of maximum horizontal stress, $S_{\mathrm{Hmax}}$, with standard deviation sourced from hydrofracturing tests in coal mines (areas shaded in grey); LRP - Lower Rhine Plain, WL - Westphalian Lowlands, RM - Rhenish Massif. The rose plots show the fault strike values considered in this study (in grey) and $S_{\mathrm{Hmax}}$ azimuth indicators from the region (Kruszewski et al., 2022a) (in green). The cross-sections from b) marked with red dashed lines. The Digital Elevation Model is from USGS (2018). b) Simplified geological cross-sections of the region (after Jansen et al. 1986, Drozdzewski et al. 2007, and Balcewicz et al. 2021).





**Figure 2.** Histograms of input variables used for calculation of slip tendency, $T_s$, of the NW-SE-striking fault set from distributions presented in Table 1 with mean or mode values represented with a solid red vertical line: a) distribution of the vertical stress, $S_v$; b) distribution of maximum horizontal stress, $S_{Hmax}$; c) distribution of minimum horizontal stress, $S_{hmin}$; d) distribution of $S_{Hmax}$ azimut; e) distribution of fault strike; f) distribution of fault dip angle; g) distribution of fluid (pore) pressure, $P_f$.



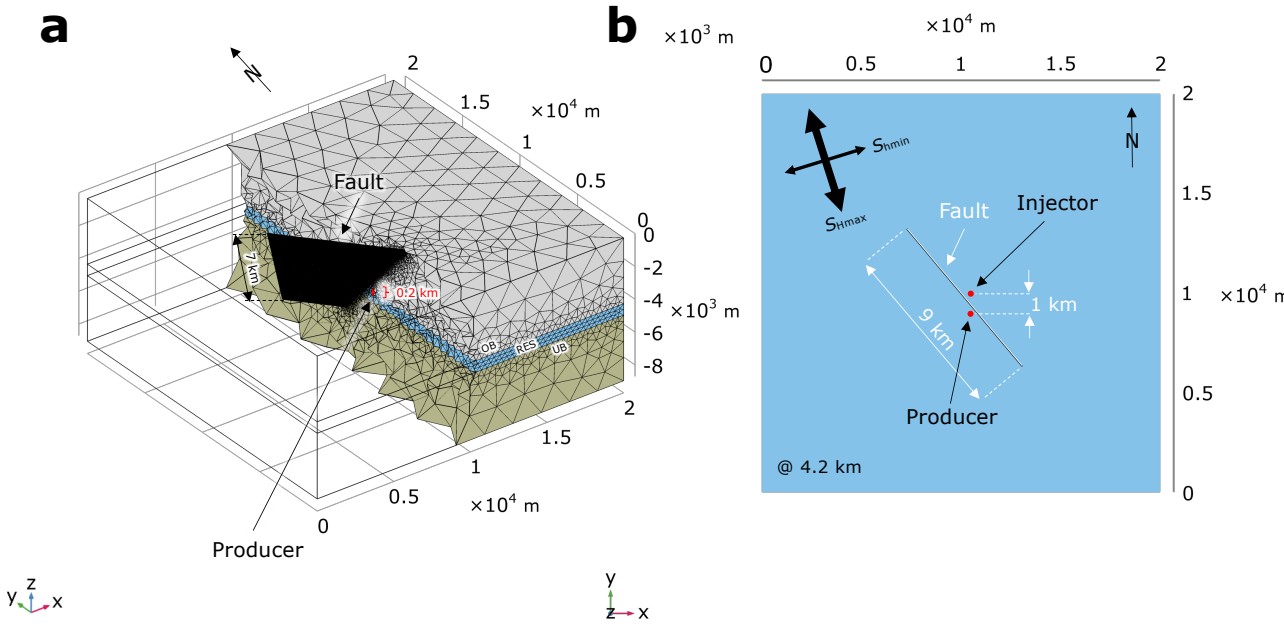

**Figure 3.** Model geometry with a NW-SE-striking fault: a) discretization of the model geometry with three geological layers i.e., overburden (OB), reservoir (RES), and underburden (UB) and a fault; injection and production wells are marked in red (injection well is hidden behind the fault). The numerical model volume was discretised into approximately 730.000 elements; b) cross-section at depth of 4.2 km (i.e., reservoir depth) with the geometry of the fault zone and injection and production wells marked with red points. Fault strike is 140°, dip angle is 60°, and initial slip tendency, $T_s$, is 0.56.





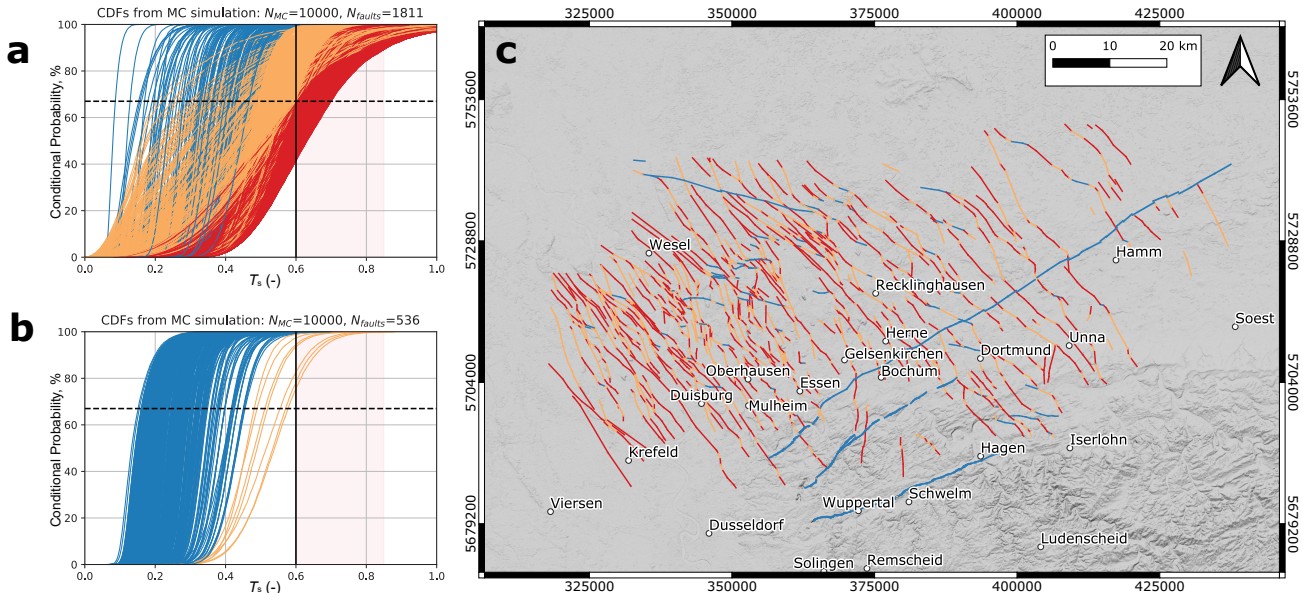

**Figure 4.** Results of the slip tendency, $T_s$, analysis of faults in the greater Ruhr region based on the analytical-probabilistic model: a) cumulative density function (*CDF*) plot for the set of NW-SE-striking faults (*n* = 1811). Red colour represents fault segments with >33 % chance of exceeding threshold friction, $\mu$, of 0.6 (solid black vertical line), amber shows between >1% and <33% chance, whereas blue shows <1% chance. The range of possible $\mu$ (i.e., between 0.6 and 0.85) is shown with pink shading; b) *CDF* plot for the set of NE-SW-striking faults (*n* = 536); c) a map view of all major faults in the greater Ruhr region coloured in accordance to probabilities from a) and b). The Digital Elevation Model is from USGS (2018).



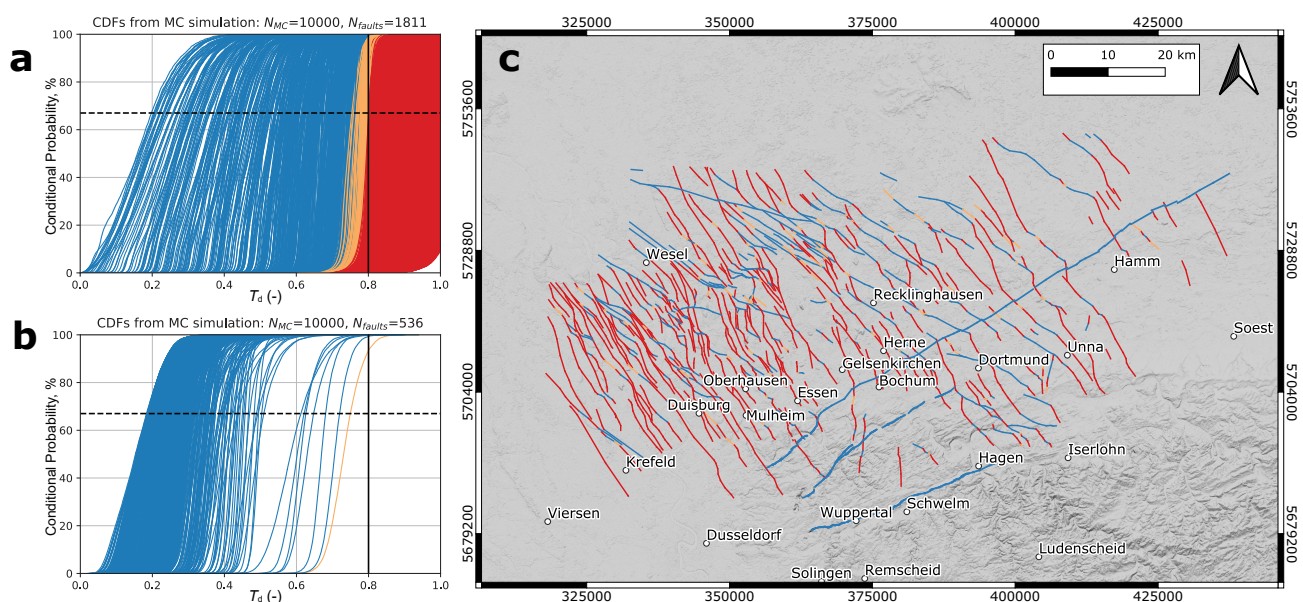

**Figure 5.** Results of dilation tendency, $T_d$, analysis for major faults in the greater Ruhr region based on the analytical-probabilistic model: a) cumulative density function (*CDF*) plot for the set of NW-SE-striking faults (*n* = 1811). Red colour represents fault segments with >33 % chance of exceeding threshold $T_d$ of 0.8 (solid black vertical line), amber shows between >1% and <33% chance, whereas blue shows <1% chance; b) *CDF* plot for the set of NE-SW-striking faults (*n* = 536); c) a map view of all major faults in the greater Ruhr region coloured in accordance to probabilities from a) and b). The Digital Elevation Model is from USGS (2018).



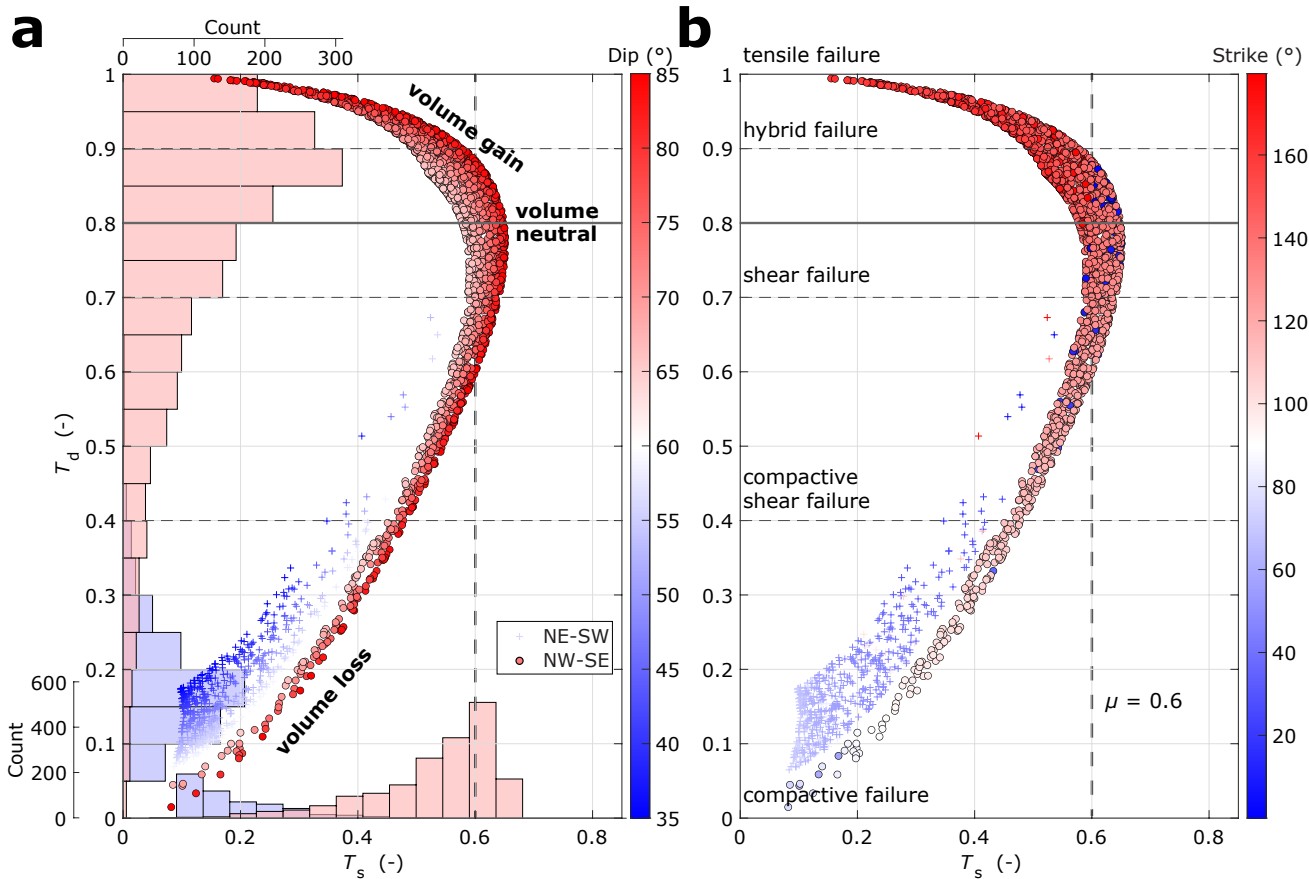

**Figure 6.** Dilation tendency, $T_d$, in a function of slip, $T_s$, tendency for all fault segments used in this study ($n = 2347$) with failure/reactivation modes (dashed solid horizontal black lines) and different marker colors representing a) fault dip angles and b) fault strike values. Red and blue bars represent histograms of the NW-SE- and NE-SW-striking fault segments, respectively. Dashed solid vertical black line represents the threshold sliding friction coefficient, $\mu$ of 0.6, whereas the thick solid black line, indicates $T_d$ threshold of 0.8.



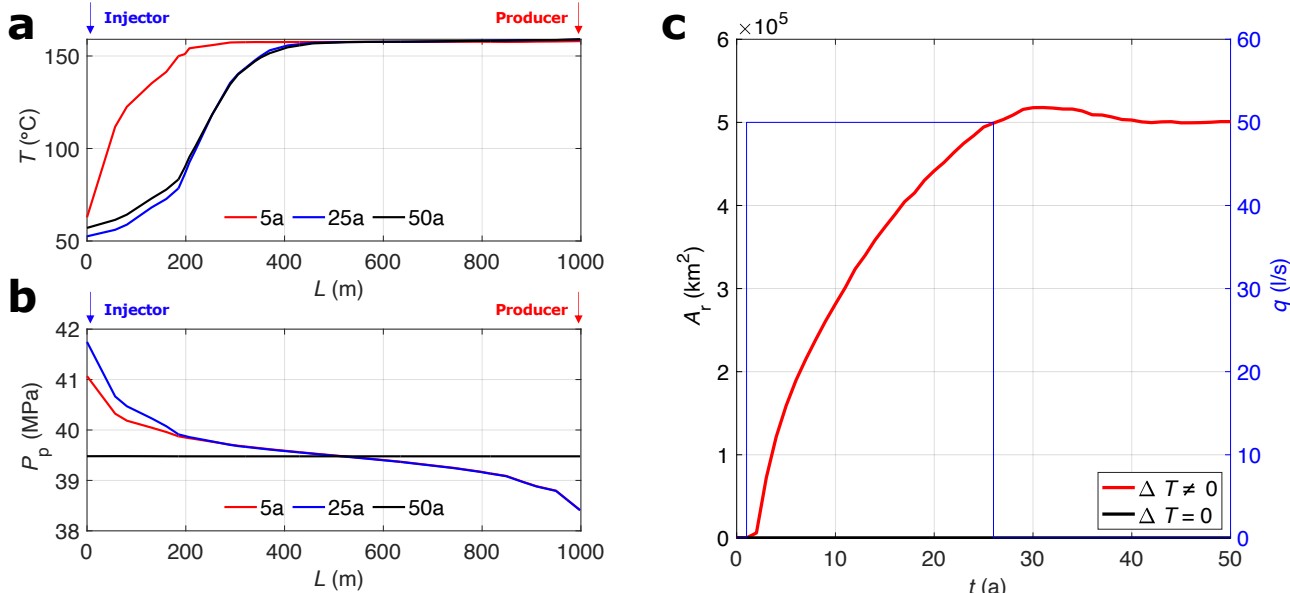

**Figure 7.** a) Evolution of temperature, $T$, and b) pore pressure, $P_p$, along the shortest distance, $L$, between injection and production well at a depth of 4.2 km for a nonisothermal coupled numerical model with NW-SE-striking fault at different simulation times; c) cumulative reactivated fault area, $A_r$, (i.e., fault segments with $T_s \geq 0.6$) fault for isothermal (black) and nonistohermal (red) numerical models with the injection and production flow rates, $q$, represented with blue line.

**Table 1.** Distribution of input parameters used for the analytical-probabilistic model at a reservoir depth of 4.5 km ($\bar{x}$ - mean value; $SD$ - standard deviation; $\kappa$ - measure of concentration; $Kurt$ - kurtosis; $P_p$ - pore pressure; $S_{hmin}$ - minimum horizontal stress magnitude, $S_{Hmax}$ - maximum horizontal stress magnitude, $S_v$ - vertical stress magnitude, $\mu$ - coefficient of sliding friction, $C_0$ - cohesion;). *Hydrostatic pressure with fluid density of 1000 kg m$^{-3}$; **Extracted from geological maps.

| Property | $\bar{x}$ | $SD$, $\kappa$ | Unit | Distribution | Reference |
|---|---|---|---|---|---|
| $P_p$* | 44 | 2.2 (5% of mean) | MPa | Gaussian | - |
| $S_{hmin}$ | 65 | 6.5 (10% of mean) | MPa | Skewed normal ($Kurt$ = +3.5) | Kruszewski et al. (2022a) |
| $S_{Hmax}$ | 114 | 26 (22.5% of mean) | MPa | Skewed normal ($Kurt$ = +2.9) | Kruszewski et al. (2022a) |
| $S_v$ | 111 | 6 (5% of mean) | MPa | Gaussian | Kruszewski et al. (2022a) |
| Azimuth of $S_{Hmax}$ | 161 | 30 | ° | Von Mises (circular normal) | Kruszewski et al. (2022a); Heidbach et al. (2018) |
| Strike** | - | - | ° | - | GD NRW (2017, 2019) |
| Dip (NW-SE) | 65 - 85 | - | ° | Uniform | Jansen et al. (1986); Drozdzewski et al. (2007) |
| Dip (NE-SE) | 35 - 60 | - | ° | Uniform | Jansen et al. (1986); Drozdzewski et al. (2007) |
| $\mu$ | 0.6 | 0.1 (10% of mean) | - | Gaussian | Byerlee (1978) |
| $C_0$ | 1 | 0.2 (20% of mean) | MPa | Gaussian | Wyllie and Norrish (1996) |



**Figure 8.** Spatial changes of shear stress, $\tau$, effective normal stress, $\sigma_n$', and slip tendency, $T_s$, on a NW-SE-striking fault with strike of $140°$, dip angle of $70°$, and an initial $T_s$ of 0.57 in the greater Ruhr region presented for the case of $5^{th}$ (a, d, g), $25^{th}$ (b, e, h), and $50^{th}$ (c, f, i) simulation year (*OB* - overburden; *RES* - reservoir; *UB* - underburden); Area of the fault with $T_s \geq 0.6$ is marked with a black outline in g), h), and i).



**Figure 9.** Results of the reduced-risk dilation tendency, $T_{dn}$, analysis for all major faults in the greater Ruhr region ($n = 2347$). The stereonet of $T_{dn}$ for any arbitrarily oriented fault in the greater Ruhr region is located in the top left corner of the map. The Digital Elevation Model is from USGS (2018).





**Table 2.** Parameters used for the coupled thermo-hydro-mechanical numerical simulations, where $d$ - thickness; $p_B$ - bulk rock density; $E_s$ - static Young's modulus; $\nu$ - Poisson's ratio; $\beta$ - Biot coefficient; $k$ - matrix permeability; $\phi$ - porosity; $\lambda$ - thermal conductivity; $\alpha$ - coefficient of thermal expansion; $C_p$ - heat capacity at constant pressure; [a]Drozdzewski et al. (2007), [b]Jansen et al. (1986), [c]DEKORP (1990), [d]Farkas et al. (2021), [e]RUBITEC (2003), [f]Brenne (2016), [g]Lippert et al. (2022), [h]Manger et al. (1963), [i]Balcewicz et al. (2021), [j]Mullen et al. (2007), [k]Robertson (1988), [l]Griffith (1936).

| Property | Overburden | Reservoir | Underburden | Fault |
|---|---|---|---|---|
| $d$ (m) | [a, b, c]4000 | [a, b, c]700 | [a, b, c]4000 | [d]1E-4 |
| $\rho_B$ (kg m$^{-3}$) | [e, f]2500 | [g]2604 | [h]2504 | 2000 |
| $E_s$ (MPa) | [e]52.5 | [i, j]50.0 | [f]29.0 | - |
| $\nu$ (-) | [e]0.25 | [i]0.33 | [f]0.17 | - |
| $\beta$ (-) | 1 | 1 | 1 | - |
| $k$ (m$^2$) | [e]1E-18 | 1E-14 | [e]1E-18 | 1E-11 |
| $\phi$ (-) | [e]0.01 | [g]0.04 | [f]0.02 | 0.50 |
| $\lambda$ (W m$^{-1}$K$^{-1}$) | [e]3.6 | [g]3.1 | [f]3.5 | 3.0 |
| $C_p$ (J kg$^{-1}$K$^{-1}$) | [e]818 | [k]880 | [k]880 | 800 |
| $\alpha$ (K$^{-1}$) | [l]1.5E-5 | [k]8E-6 | [l]1.5E-5 | - |



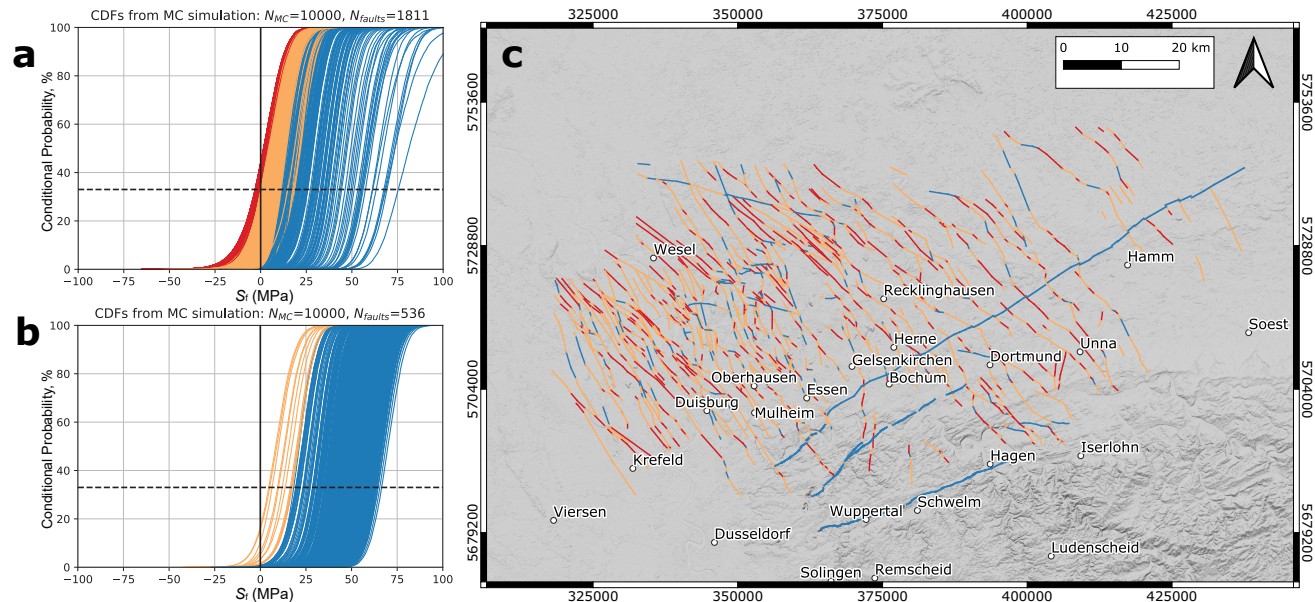

**Figure A1.** The results of fracture susceptibility, $S_f$, for major faults in the greater Ruhr region based on the analytical-probabilistic model: a) cumulative density function (*CDF*) plot for the set of NW-SE-striking faults ($n = 1811$). Red colour represents fault segments with >33 % chance of exceeding threshold $S_f$ of 0 MPa (solid black vertical line), amber shows between >1% and <33% chance, whereas blue shows <1% chance; b) *CDF* plot for the set of NE-SW-striking faults ($n = 536$); c) a map view of all major faults in the greater Ruhr region coloured in accordance to probabilities from a) and b). The Digital Elevation Model was sourced from USGS (2018).





**Figure A2.** Spatial changes of shear stress, $\tau$, effective normal stress, $\sigma_n$', and slip tendency, $T_s$, on a NE-SW-striking fault with strike of 70° and dip angle of 50° and an initial $T_s$ of 0.18 in the greater Ruhr region presented for the case of 5th (a, d, g), 25th (b, e, h), and 50th (c, f, i) simulation year (*OB* - overburden; *RES* - reservoir; *UB* - underburden).