# Peer review of "Evolution of fault reactivation potential in deep geothermal systems. Insights from the greater Ruhr region, Germany"

_EGUsphere, 2023_

## Referee Comment (RC2)

[revised manuscript text omitted]
_v$, minimum horizontal, $S_{hmin}$, and maximum horizontal, $S_{Hmax}$, stresses, the azimuth of $S_{Hmax}$, and $P_p$. Two additional input parameters were used for the case of $S_f$ including $\mu$ and $C_0$. The input data used for the analytical-probabilistic model were extrapolated at the assumed reservoir depth based on the available geological data from the study region sourced from the Carboniferous layers (Table 1). We compute $T_s$, $T_d$, and $S_f$ for a reservoir depth of $4.5\,\mathrm{km}$, i.e., depth of Devonian carbonates (DEKORP, 1990), separately for a set of NW-SE- and NE-SW-striking fault segments. We performed 10,000 simulations for
130   each fault segment, of the three scalar values and plotted the resultant cumulative distribution functions, *CDF*.

Each computed scalar value, $T_s$, $T_d$, and $S_f$, has a corresponding threshold, i.e., a critical value. In the case of $T_s$, we define a threshold value of 0.6 (Byerlee, 1978), which assumes fault reactivation for values of $T_s$ that are greater or equal to the threshold. For $T_d$, we assume a threshold value of 0.8, based on a case study by Ferrill et al. (2020a), which shows that mapped fractures with $T_d$ exceeding 0.8 exhibit mineral coating, being indicative of crystal growth in an open void experiencing
135   dilation. The open void scenario is in contrast with slickenlined surfaces from the same study with $T_d$ below 0.8 with negligible observed dilation. We assume a threshold of $0\,\mathrm{MPa}$ for the case of $S_f$ which suggests that faults experiencing negative $S_f$ values are unstable and subject to reactivation when subjected to insignificantly small stress changes. Stable faults, on the other hand, experience positive $S_f$ values.

We extracted fault strike values from an open access map of major faults in the greater Ruhr region (GD NRW, 2017, 2019).
140   The data set include a total of 2347 fault segments, with a combined fault length of $2596\,\mathrm{km}$. For each fault segment, we selected a random dip angle based on a uniform distribution for both NW-SE- and NE-SW-striking faults with an assumption of a dip angle being between $65°$ and $85°$ for the former and between $35°$ and $60°$ for the latter. We used such distribution to account for the high uncertainty of fault geometry at the reservoir depth, and to represent all dip angles known from the available geological maps in the region (Jansen et al., 1986; Drozdzewski et al., 2007).

145   We used an *in situ* stress database from the greater Ruhr region based on an high number of 429 hydrofracturing tests to constrain distributions of $S_{hmin}$, $S_{Hmax}$, $S_v$, and $S_{Hmax}$ azimuth at the reservoir depth of $4.5\,\mathrm{km}$. Using the aforementioned data, we applied a skewed data distribution for both $S_{hmin}$ and $S_{Hmax}$ magnitudes, Gaussian distribution for $S_v$ magnitude, and

[Figure]

[Figure]

Von Misses (circular) distribution for $S_{\text{Hmax}}$ azimuth. We assume $P_{\text{
[revised manuscript text omitted]
_\mathrm{p}$ - pore pressure; $S_\mathrm{hmin}$ - minimum horizontal stress magnitude, $S_\mathrm{Hmax}$ - maximum horizontal stress magnitude, $S_\mathrm{v}$ - vertical stress magnitude, $\mu$ - coefficient of sliding friction, $C_0$ - cohesion;). *Hydrostatic pressure with fluid density of 1000 kg m$^{-3}$; **Extracted from geological maps.

| Property | $\bar{x}$ | $SD$, $\kappa$ | Unit | Distribution | Reference |
|---|---|---|---|---|---|
| $P_\mathrm{p}$* | 44 | 2.2 (5% of mean) | MPa | Gaussian | - |
| $S_\mathrm{hmin}$ | 65 | 6.5 (10% of mean) | MPa | Skewed normal ($Kurt = +3.5$) | Kruszewski et al. (2022a) |
| $S_\mathrm{Hmax}$ | 114 | 26 (22.5% of mean) | MPa | Skewed normal ($Kurt = +2.9$) | Kruszewski et al. (2022a) |
| $S_\mathrm{v}$ | 111 | 6 (5% of mean) | MPa | Gaussian | Kruszewski et al. (2022a) |
| Azimuth of $S_\mathrm{Hmax}$ | 161 | 30 | ° | Von Mises (circular normal) | Kruszewski et al. (2022a); Heidbach et al. (2018) |
| Strike** | - | - | ° | - | GD NRW (2017, 2019) |
| Dip (NW-SE) | 65 - 85 | - | ° | Uniform | Jansen et al. (1986); Drozdzewski et al. (2007) |
| Dip (NE-SE) | 35 - 60 | - | ° | Uniform | Jansen et al. (1986); Drozdzewski et al. (2007) |
| $\mu$ | 0.6 | 0.1 (10% of mean) | - | Gaussian | Byerlee (1978) |
| $C_0$ | 1 | 0.2 (20% of mean) | MPa | Gaussian | Wyllie and Norrish (1996) |

[Figure]

**Figure 8.** Spatial changes of shear stress, $\tau$, effective normal stress, $\sigma_n$', and slip tendency, $T_s$, on a NW-SE-striking fault with strike of 140°, dip angle of 70°, and an initial $T_s$ of 0.57 in the greater Ruhr region presented for the case of 5[th] (a, d, g), 25[th] (b, e, h), and 50[th] (c, f, i) simulation year (*OB* - overburden; *RES* - reservoir; *UB* - underburden); Area of the fault with $T_s \geq 0.6$ is marked with a black outline in g), h), and i).

[Figure]

[Figure]

**Figure 9.** Results of the reduced-risk dilation tendency, $T_{dn}$, analysis for all major faults in the greater Ruhr region ($n = 2347$). The stereonet of $T_{dn}$ for any arbitrarily oriented fault in the greater Ruhr region is located in the top left corner of the map. The Digital Elevation Model is from USGS (2018).

[Figure]

**Table 2.** Parameters used for the coupled thermo-hydro-mechanical numerical simulations, where $d$ - thickness; $p_B$ - bulk rock density; $E_s$ - static Young's modulus; $\nu$ - Poisson's ratio; $\beta$ - Biot coefficient; $k$ - matrix permeability; $\phi$ - porosity; $\lambda$ - thermal conductivity; $\alpha$ - coefficient of thermal expansion; $C_p$ - heat capacity at constant pressure; [a]Drozdzewski et al. (2007), [b]Jansen et al. (1986), [c]DEKORP (1990), [d]Farkas et al. (2021), [e]RUBITEC (2003), [f]Brenne (2016), [g]Lippert et al. (2022), [h]Manger et al. (1963), [i]Balcewicz et al. (2021), [j]Mullen et al. (2007), [k]Robertson (1988), [l]Griffith (1936).

| Property | Overburden | Reservoir | Underburden | Fault |
|:---:|:---:|:---:|:---:|:---:|
| $d$ (m) | [a,b,c]4000 | [a,b,c]700 | [a,b,c]4000 | [d]1E-4 |
| $\rho_B$ (kg m$^{-3}$) | [e,f]2500 | [g]2604 | [h]2504 | 2000 |
| $E_s$ (MPa) | [e]52.5 | [i,j]50.0 | [f]29.0 | - |
| $\nu$ (-) | [e]0.25 | [i]0.33 | [f]0.17 | - |
| $\beta$ (-) | 1 | 1 | 1 | - |
| $k$ (m$^2$) | [e]1E-18 | 1E-14 | [e]1E-18 | 1E-11 |
| $\phi$ (-) | [e]0.01 | [g]0.04 | [f]0.02 | 0.50 |
| $\lambda$ (W m$^{-1}$K$^{-1}$) | [e]3.6 | [g]3.1 | [f]3.5 | 3.0 |
| $C_p$ (J kg$^{-1}$K$^{-1}$) | [e]818 | [k]880 | [k]880 | 800 |
| $\alpha$ (K$^{-1}$) | [l]1.5E-5 | [k]8E-6 | [l]1.5E-5 | - |

[Figure]

[Figure]

**Figure A1.** The results of fracture susceptibility, $S_f$, for major faults in the greater Ruhr region based on the analytical-probabilistic model: a) cumulative density function (*CDF*) plot for the set of NW-SE-striking faults (*n* = 1811). Red colour represents fault segments with >33 % chance of exceeding threshold $S_f$ of 0 MPa (solid black vertical line), amber shows between >1% and <33% chance, whereas blue shows <1% chance; b) *CDF* plot for the set of NE-SW-striking faults (*n* = 536); c) a map view of all major faults in the greater Ruhr region coloured in accordance to probabilities from a) and b). The Digital Elevation Model was sourced from USGS (2018).

[Figure]

[Figure]

[Figure]

**Figure A2.** Spatial changes of shear stress, $\tau$, effective normal stress, $\sigma_n$', and slip tendency, $T_s$, on a NE-SW-striking fault with strike of 70° and dip angle of 50° and an initial $T_s$ of 0.18 in the greater Ruhr region presented for the case of 5th (a, d, g), 25th (b, e, h), and 50th (c, f, i) simulation year (*OB* - overburden; *RES* - reservoir; *UB* - underburden).

---

## Author Comment (AC1)

Dear Reviewer, we are very grateful for the careful revision of our manuscript and for making our study more complete. We benefited greatly from your feedback and, after careful revision and implementation of your comments and critiques, we are coming back with the manuscript revision and detailed replies to your comments. Please let us know in case of any questions or concerns regarding the new version of the manuscript or the replies below. Our replies are in red and your comments are in black.

I would like to emphasize that the boundary conditions and physical properties of the numerical models need to be more thoroughly explained, and the consequences for the results presented more thoroughly discussed.

We now discuss the boundary conditions in greater detail (Section 5.4) as well as show them directly on the model in Figure 8. We have changed the structure of the manuscript, where numerical modelling is now only rather small part of the discussion. We shift the main focus of the study to the probabilistic analysis of fault reactivation as well as to the analysis of the mapped faults in the greater Ruhr region within the slip tendency vs. dilation tendency parameter space.

'Ln 1 : The initial sentence should be tempered. The success of deep geothermal systems significantly relies on the integration of various geological and physical processes, which could be defined in numerical modelling by a complete THMC coupling (Chester and Logan 1986; Byerlee, 1994; Barton et al., 1995; Scholz, 2002; Violay et al., 2017, ...). Furthermore, many other parameters such as cost drilling and feed in tariffs supporting the development of geothermal energy must be taken into account to evaluate the feasibility and viability of the project.'

We agree with the reviewer. We delete this sentence from the manuscript altogether and make appropriate changes to the manuscript (see Introduction).

'Ln 2 : The terms "fault" and "fault zone" are used successively between the first and second sentences. It might be clearer for the reader to use a single term, which could be defined in the introduction.'

We accept the suggestion and use 'fault' throughout the manuscript. We now explain what we consider a 'fault' in the introduction part of the manuscript (L20-21).

'Ln 18 : I would suggest adding references here. It might also be interesting for the reader to see, at this point, the definition of "fault" that as used throughout the remainder of the study.'

We accept the comment. Reference is added and the definition of a 'fault' applicable for the manuscript is included (L20-21).

'Ln 21 : Might use a less global term than "Anthropogenic", perhaps geothermal activities? This would require a revision of the sentence structure.'

We amend the sentence as suggested by the reviewer (L23)

'Ln 24 : I suggest revising the sentence structure, perhaps changing "On the other hand" to "Moreover"?'

We amend the sentence as suggested by the reviewer (L25)

'Ln 26 : May I suggest some recent studies on this topic : Guillou-Frottier et al., 2013, Moeck, 2014, Duwiquet et al., 2019.'

References are added (L27-28).

'Ln 55 & 60 : Though comprehensive and clear thus far, your introduction might benefit from incorporating a review of the current state of the art concerning the application of these two geomechanical criteria (Ts, Td) in analogous contexts. Consider referencing studies like Moeck et al. (2009), among potentially others. The same consideration applies to THM numerical modeling. Providing a logical justification for their utilization, accompanied by a review of prior studies on this subject, would enhance the overall contextual understanding. I have in mind Armandine Les Landes et al., 2019, and/or Duwiquet et al., 2021, but undoubtedly, there are other relevant studies as well.'

We agree with the reviewer and add missing references both to the Ts, Td analysis and for the numerical modelling part (L56-71).

'Subsection 3.2: Here, you are directed towards the use of commercial software. At this stage, it might be prudent to conduct calibration tests for the employed THM coupling in comparison to the open-source codes that have been used and published.

We believe that the comparison of our numerical model developed using commercial software such as COMSOL software with other open source codes goes beyond the scope of this paper. The use of COMSOL software is a common practice in science seen already in many peer-reviewed publications e.g., *Taillefer et al., 2018* or *Kruszewski et al., 2023*. The main scope of our paper is not to investigate different finite element implementations in commercial software and open source codes. This should be done in the course of a benchmark study that uses a defined model set up for which maybe even an analytical solution exists. Our focus is to investigate the reactivation potential of faults based on probabilistic approaches using an example of the greater Ruhr region. We, anyway, publish numerical model, including all necessary information about input parameters, boundary conditions, discretization etc., with the manuscript to allow reproducibility of the results presented in our study. These can be checked by the readers of our study (as well as reviewers) in terms of their comparability with any other open source codes.

It would be advisable to directly specify the boundary conditions of the employed numerical models on Figure 3. The clarity of the figure would be enhanced by clearly displaying the dimensions of the considered system. (…)

The dimensions of the model geometry and fault geometry as well as boundary conditions are now presented in Figure 8. The details on the boundary conditions are also described in detail in section 5.4 of the manuscript (L299-307). Dimensions of the model are mentioned in the caption of Figure 8 and can be also directly deducted based on the axes in Figure 8.

(…) In the figure description, you provide information about the number of cells used, but the cell sizes vary between the fault and other lithologies. What are the minimum and maximum sizes, and how were these sizes chosen? (…)

We have now refined our numerical resolution in the presented model. The mesh size, therefore, changed from the initial model published with this manuscript. The updated model has now 1,276,857 (tetrahedral) mesh elements and 214,326 mesh vertices with average element quality of 0.66. The updated model will be made available with the revised version of the manuscript. Following minimum and maximum cell size were, therefore, selected for discretization in the updated model:

Underburden: minimum element size of 42 m and maximum of 2360 m;

Overburden: minimum element size of 37 m and maximum of 2380 m;

Reservoir: minimum element size of 4.4 m and maximum of 555 m;

Fault: minimum element size of 33 m and maximum of 66 m (we use finer mesh on the fault that is represented in the model as a plane);

Boreholes: maximum element size was limited to 5 m (we use finer mesh around the boreholes that are represented in the model as line/edge elements).

Below we include two snapshots showing the mesh size for the whole model (only volumes visible) with the maximum and minimum element sizes as well as a close-up of the mesh size around the two boreholes and the fault. The selection of mesh size was based i) on the type of physics (i.e., solid mechanics, fluid flow and heat transfer in porous media) used in the numerical model as well as ii) to achieve computational efficiency (i.e., having the same results but with shortest computational time).

[Figure]

[Figure]

(…) Have convergence tests been conducted to ensure that the numerical results are no longer dependent on the cell size? (…)

Yes, we have carried out convergence tests on models three different mesh sizes. Below a snapshot comparing results of the cumulative reactivated fault area (Ar; on the left) as well as the maximum dilation tendency on the fault plane (on the right) computed with the model discretized into $0.73 \cdot 10^6$ elements and one discretized in $1.28 \cdot 10^6$ elements. We skip the model with $0.3 \cdot 10^6$ elements, where model results were deemed to be dependent on mesh size. The results we deemed to be satisfactory and we use the model discretized in $1.28 \cdot 10^6$ elements for our discussion in the manuscript.

[Figure]

(…) Additionally, how are the meshes considered in the modeled wells? Is it a radial mesh? What impact does this have on the final result ?'

There is an ever-present difficulty in modelling small scale elements like boreholes with a diameter of tens of centimeters embedded in reservoir models spanning tens of kilometers. For computation efficiency, we decided to use 1D line/edge elements to model the two boreholes. We limit the maximum element size to 5 m; i.e., we make the mesh size in the reservoir and around boreholes finer.

'Ln 181 : It may be necessary to reconsider the dimensions of the fault (a width of 9 km?) and provide information on the fault thickness.'

The fault (along-dip) length and (along-strike) width were both based on geological information from the region, being it seismic lines (DEKORP 1990) and fault trace maps (GD 2014 and GD 2019). The mentioned fault width is actually the along-strike-width of the fault. Faults of such widths are common in the region, what can be seen in Figure 1 in the manuscript. Fault (hydraulic) thickness is mentioned and referenced in Table 2.

'Table 2 : The permeability of the fault ($10^{-11}$ m²) is not referenced. Such a high permeability value could lead to fluid flow velocities that exceed the limits of the applicability of Darcy's law. It is essential to verify this by examining whether fluid flow velocities are consistent. For example, in comparison, other numerical models (but same software) impose fault permeability values close to $10^{-14}$ m² and find corresponding field data for this value (Roche et al., 2018; Taillefer et al., 2017). It seems important here to provide further explanation of your approach. Additionally, the imposed permeability value for the reservoir also lacks a reference.'

We agree with the reviewer and carry out numerical simulations for fault permeability of 10-14 m² in and use this permeability as our fault permeability in the updated numerical model. We have tested our approaches with both permeability values and have seen rather negligible change for slip tendency or dilation tendency. This have been also discussed in more detailed in *Kruszewski et al. (2023)*. We also add reference to fault permeability of 10-14 m² in Table 2. We add a reference for reservoir permeability to Table 2 that is based on the lower limit for successful exploitation of hydrothermal reservoirs in Bavaria (*Fritzer et al. 2012*) and indication of reservoir permeability from the Californie geothermal field in the western Netherlands in the Lower Rhine Graben (based on the technical reports of the *SCAN* project; *https://www.nlog.nl/en/scan*).

'Ln 200 : You fixed a temperature at the bottom of the model. In order to limit boundary conditions at the model's base, wouldn't it be preferable to use a heat flux instead? These aspects could be either modified or discussed in the relevant section.'

We set the following values of temperature at the top and bottom sides of the model to merely recreate a temperature gradient across the numerical model based on the geothermal gradient of the region of 35 °C/km (reference included in the manuscript). The initial temperature field in the model represents, therefore, the geothermal gradient of the region. We find this assumption to be the most suitable solution for sedimentary regions like the Ruhr region where no thermal anomalies are expected. Our model does not intent to model groundwater movement, and connected temperature changes due to faulting, in and around the reservoir. As a result, we do not see the necessity for using the heat flux as boundary conditions and are convinced that the approach we used is sufficient for investigating the normal and

shear stresses and resultant slip and dilation tendencies on faults. We amend the manuscript text to include abovementioned points (L305-306).

---

## Author Comment (AC2)

Dear Reviewer, we are very grateful for the careful revision of our manuscript and for making our study more complete. We benefited greatly from your feedback and, after careful revision and implementation of your comments and critiques, we are coming back with the manuscript revision and detailed replies to your comments. Please let us know in case of any questions or concerns regarding the new version of the manuscript or the replies below. Our replies are in red and your comments are in black.

The rationale for having the two types of analyses, static and numerical continuum, is not clear to me as insights from the first do not strongly pertain to the second. The analyses do agree with each other and support the findings and the discussion so I propose that the linkage between the two be strengthened.

We have decided to apply a different approach in the revised version of our study. We amended the structure of the manuscript focusing on the uncertainty analysis using probabilistic approaches as well as the Ts vs. Td parameter space for location of prospecting of structurally controlled geothermal systems. The use of numerical modelling is decreased significantly and is now included only in the discussion part as a support for the arguments stated in the study.

The manuscript is written fairly clearly but suffers significantly from being highly repetitive (e.g., the text in the appendix is also presented in the main text). It is repetitive in many other aspects.

The repetitiveness has been corrected. Text from the appendix was integrated within the main parts of the manuscript. Other repetitions across the manuscript were removed or kept to the minimum.

I recommend that the manuscript be thoroughly rewritten as it is considered for publication. Unless a figure/table maximum prohibits, the figures in the appendix can be moved to the main text.

We decided to remove some figures from the manuscript focusing more on the probabilistic analysis of mapped faults rather than numerical modelling. One figure (now Figure 5) from the appendix was moved to the main text.

The manuscript deals with static analysis of faults and coupled simulation of a faulted continuum with the main application being the likelihood of faults reactivation. There is no treatment of seismicity (e.g., moments, maximum magnitude, slow vs fast rupture, rate and state, etc). Therefore, mention of application to seismicity should be kept to a minimum with the focus kept on reactivation.

We agree with this comment and include appropriate changes to the manuscript. It should not be, however, forgotten that higher values of slip tendency are related to a higher likelihood for fault reactivation which could lead to higher probability of induced seismic events during geothermal production/circulation. We acknowledge, however, that more data and studies are needed to understand the relationship between slip tendency and size of an earthquake. This has been also discussed in the manuscript at the end of the section 5.4 (L328-336).

As specifically written, equation 3 is not in Streit and Hills (2004) therefore some additional description is required.

We add additional references and explain that the equation is explicitly for a case of a cohesionless fault (L132-133).

Cohesion appears in some of the analysis but not consistently.  Either don't use it and discuss why, or use it throughout.

We remove cohesion from the analysis and from the manuscript focusing on an idealized case of a cohesionless fault(s). As a result, we amend calculations of Sf and include these changes in the manuscript. Due to this, we provide new maps of Sf in the revised version of this manuscript.

"distance to failure" is colloquial wording.  Replace with the technical equivalent pertaining to stress condition relative to failure condition.

We remove this phrasing from the manuscript.

**A marked pdf with many suggestions for rewording and clarification has been provided.**
**Citation**: https://doi.org/10.5194/egusphere-2023-1889-RC2

Other suggestions provided with a .pdf file were integrated into the manuscript. We also attach a .pdf file with all detailed replies to the reviewer comments.

[Figure]

[Figure]

**Evolution of fault reactivation potential in deep geothermal systems. Insights from the greater Ruhr region, Germany**

Michal Kruszewski[1], Alessandro Verdecchia[2], Oliver Heidbach[3,4], Rebecca M. Harrington[2], and David Healy[5]

[1]Chair of Engineering Geology and Hydrogeology, RWTH Aachen University, Lochnerstraße 4-20, 52056 Aachen, Germany
[2]Institute of Geology, Mineralogy, and Geophysics, Ruhr-University Bochum, Universitätsstraße 150, 44801 Bochum, Germany
[3]GFZ German Research Centre for Geosciences, Telegrafenberg, 14473 Potsdam, Germany
[4]Institute for Applied Geosciences, Technical University Berlin, Ernst-Reuter Platz 1, 10587 Berlin, Germany
[5]Department of Geology & Geophysics, School of Geosciences, University of Aberdeen, Aberdeen AB24 3UE, United Kingdom

**Correspondence:** Michal Kruszewski (kruszewski@lih.rwth-aachen.de)

**Abstract.** The success of deep geothermal systems depends on the presence of fault zones in the subsurface. Faults play a vital role in the Earth's plumbing system by facilitating fluid flow when they dilate, but are simultaneously known to enhance the hazard of the system once slipping in shear mode. As dilation of a fault enhances its permeability significantly, shear failure can lead to loss of boreholes or seismic events of  concern. In this study, we present the evolution of reactivation

5    potential of major faults during 25-year production period in deep generic geothermal systems in the greater Ruhr region in western Germany. To determine the pre-operational *in situ* stress state we use a recently published comprehensive dataset of stress magnitude data from the greater Ruhr region in an analytical-probabilistic model accounting for uncertainties of *in situ* stress, fault geometry, and frictional properties for a prospective reservoir in the Devonian *Massenkalk* formations. The resulting cumulative distribution functions of dilation and slip tendency of given fault sets suggests that more than half of

10   the combined length of NW-SE-striking faults have a high reactivation probability, whereas the NE-SW-striking faults remain not optimally-oriented in the regional stress field. Using the relationship between dilation and slip tendency, we propose fault segments suitable for geothermal development that exhibit high hydraulic conductivity, i.e. high dilation tendency, and lower potential for shear failure, i.e. low slip tendency. In the second step, we employ generic thermo-hydro-mechanical models to quantify induced spatio-temporal stress changes on selected fault planes due to long-term geothermal production. We find

15   that after 25 years thermal stress changes contribute significantly to the change of the reactivation potential which should be accounted for while planning deep geothermal systems.

**1   Introduction**

The Earth's crust is characterised by a complex web of faults, fractures, and zones of inherent weakness, shaped by millions of years of tectonic activity. Although all faults in the subsurface have the potential to be reactivated, their resistance to failure is

20   not uniform. The likelihood of fault reactivation depends on a number of factors, including the *in situ* stress conditions, fault
* * *
**Number: 1**     Author: 1 Subject: Inserted Text     Date: 15/10/2023 17:16:52

societal?

> Author: kruszewski  Subject: Sticky Note     Date: 04/05/2024 23:39:31
> we accept the suggestion

> Author: kruszewski  Subject: Sticky Note     Date: 04/05/2024 23:39:31
* * *
**Number: 2**     Author: 1 Subject: Comment on Text     Date: 15/10/2023 17:19:04

might be confusing in this context.  Perhaps use "sensitivity"?

> Author: kruszewski  Subject: Sticky Note     Date: 04/05/2024 23:40:06
> we accept the suggestion and do not use the phrase throughout the manuscript.

[revised manuscript text omitted]

**Number: 1**      Author: 1 Subject: Comment on Text     Date: 15/10/2023 17:23:36

again, without introduction of a failure model, "distance" is confusing

> Author: kruszewski   Subject: Sticky Note      Date: 04/05/2024 23:40:15
> we accept the suggestion

**Number: 2**      Author: 1 Subject: Highlight    Date: 04/05/2024 23:40:34

we accept the suggestion and make appropriate changes in the manuscript.

**Number: 3**      Author: 1 Subject: Inserted Text      Date: 15/10/2023 17:34:48

of?

> Author: kruszewski   Subject: Sticky Note      Date: 04/05/2024 23:40:39
> we accept the suggestion and make appropriate changes in the manuscript.

[revised manuscript text omitted]

We construct the analytical-probabilistic model using the updated version of the methodology developed in Healy and Hicks (2022), based on a combined Monte Carlo, response surface methodology, and Mohr-Coulomb theory, where $T_s$, $T_d$, and $S_f$ are computed for multiple fault segments simultaneously. $T_s$ defined as a ratio between the shear stress, $\tau$, and the effective normal stress, $\sigma_n$', is given by the following (Morris et al., 1996)

$$T_s = \frac{\tau}{\sigma_n - P_p} = \frac{\tau}{\sigma_n'} , \tag{1}$$

Number: 1       Author: 1 Subject: Inserted Text       Date: 15/10/2023 17:39:32

thick?

Author: kruszewski  Subject: Sticky Note       Date: 04/05/2024 23:41:06

we accept the suggestion and make appropriate changes in the manuscript.

Number: 2       Author: 1 Subject: Comment on Text       Date: 15/10/2023 17:48:39

incomplete sentence

Author: kruszewski  Subject: Sticky Note       Date: 04/05/2024 23:41:13

we accept the suggestion and make appropriate changes in the manuscript.

[Figure]

[Figure]

where, $\sigma_n$ is the normal stress exerted on a fault plane and $P_p$ is the pore pressure. $T_d$ is computed by (Ferrill et al., 1999):

$$T_d = \frac{\sigma_1' - \sigma_n'}{\sigma_1' - \sigma_3'} \, , \tag{2}$$

where, $\sigma_1'$ and $\sigma_3'$ are the maximum and minimum effective principal stresses, respectively. Finally, $S_f$, also known as critical pore pressure, $\Delta P_p$, being described as a change in pore fluid pressure needed to push given fault to failure, is defined as (Streit and Hillis, 2004)

$$S_f = \Delta P_p = \sigma_n' - \frac{\tau - C_0}{\mu} \, , \tag{3}$$

where, $C_0$ is the cohesive strength of a fault and $\mu$ is the static friction coefficient. The computed values of $T_s$, $T_d$, and $S_f$ of mapped fault segments derive from the distributions of several input parameters, including the magnitudes of vertical, $S_v$, minimum horizontal, $S_{hmin}$, and maximum horizontal, $S_{Hmax}$, stresses, the azimuth of $S_{Hmax}$, and $P_p$. Two additional input parameters were used for the case of $S_f$ including $\mu$ and $C_0$. The input data used for the analytical-probabilistic model were extrapolated at the assumed reservoir depth based on the available geological data from the study region sourced from the Carboniferous layers (Table 1). We compute $T_s$, $T_d$, and $S_f$ for a reservoir depth of 4.5 km, i.e., depth of Devonian carbonates (DEKORP, 1990), separately for a set of NW-SE- and NE-SW-striking fault segments. We performed 30,000 simulations for each fault segment, of the three scalar values and plotted the resultant cumulative distribution functions, *CDF*.

Each computed scalar value, $T_s$, $T_d$, and $S_f$, has a corresponding threshold, i.e., a critical value. In the case of $T_s$, we define a threshold value of 0.6 (Byerlee, 1978), which assumes fault reactivation for values of $T_s$ that are greater or equal to the threshold. For $T_d$, we assume a threshold value of 0.8, based on a case study by Ferrill et al. (2020a), which shows that mapped fractures with $T_d$ exceeding 0.8 exhibit mineral coating, being indicative of crystal growth in an open void experiencing dilation. The open void scenario is in contrast with slickenlined surfaces from the same study with $T_d$ below 0.8 with negligible observed dilation. We assume a threshold of 0 MPa for the case of $S_f$ which suggests that faults experiencing negative $S_f$ values are unstable and subject to reactivation when subjected to insignificantly small stress changes. Stable faults, on the other hand, experience positive $S_f$ values.

We extracted fault strike values from an open access map of major faults in the greater Ruhr region (GD NRW, 2017, 2019). The data set include a total of 2347 fault segments, with a combined fault length of 2596 km. For each fault segment, we selected a random dip angle based on a uniform distribution for both NW-SE- and NE-SW-striking faults with an assumption of a dip angle being between 65° and 85° for the former and between 35° and 60° for the latter. We used such distribution to account for the high uncertainty of fault geometry at the reservoir depth, and to represent all dip angles known from the available geological maps in the region (Jansen et al., 1986; Drozdzewski et al., 2007).

We used an *in situ* stress database from the greater Ruhr region based on an high number of 429 hydrofracturing tests to constrain distributions of $S_{hmin}$, $S_{Hmax}$, $S_v$, and $S_{Hmax}$ azimuth at the reservoir depth of 4.5 km. Using the aforementioned data, we applied a skewed data distribution for both $S_{hmin}$ and $S_{Hmax}$ magnitudes, Gaussian distribution for $S_v$ magnitude, and

**T** Number: 1        Author: 1 Subject: Comment on Text    Date: 17/10/2023 20:50:00

This is not in Streit and Hillis

    Author: kruszewski  Subject: Sticky Note       Date: 04/05/2024 23:41:26

we accept the suggestion and make appropriate changes in the manuscript.

    Author: kruszewski  Subject: Sticky Note       Date: 05/05/2024 20:40:30

We add additional references to this equation in the manuscript.

**T** Number: 2        Author: 1 Subject: Comment on Text    Date: 17/10/2023 20:48:01

this needs to be some other sort of symbol other than pore pressure change, such as critical DPp (DPpc ?)

**T** Number: 3        Author: 1 Subject: Comment on Text    Date: 15/10/2023 17:53:54

why 10,000?

    Author: kruszewski  Subject: Sticky Note       Date: 05/05/2024 20:40:01

This is common assumption for the Monte Carlo analysis in the literature (see Healy and Hicks, 2022 or Kruszewski et al., 2021). We use, however, 10000 for computational efficiency as well as robustness of the results.

[Figure]

Von Misses (circular) distribution for $S_{\text{Hmax}}$ azimuth. We assume $P_{\text{p}}$ following a Gaussian distribution and being equal to the hydrostatic pressure of a cold water column with density of $1000\,\text{kg}\,\text{m}^{-3}$. For the computation of $S_{\text{f}}$, both $\mu$ and $C_0$ were

150 assumed to follow Gaussian distributions. Table 1 shows the input parameters used for establishing data distributions, and Figure 2 shows an example of data distributions used for $T_{\text{s}}$ computation of the NW-SE-striking fault segments.

**3.2 Modelling the evolution of the distance to failure with THM model**

We define two generic settings for the development of a deep fault-based geothermal system and simulate THM processes during 25 years of geothermal production and 25 years of reservoir reaching equilibrium state without fluid circulation in a

155
oublet system. The selection of the two case scenarios was based on the two extreme cases of fault being close to critical state and a fault being far away from failure, which also mirrors the two prevailing fault orientations in the greater Ruhr region. Such analysis constitutes an investigation of spatio-temporal changes of $P_{\text{p}}$ and stress during long-term injection/production operations.We model the single-phase fluid flow in faults following Darcy's law, which are represented as 2D planes of weakness cutting the 3D poro-elastic model volume. Fluid is injected into the reservoir through an injection well, and produced from a

160 production well within a so-called geothermal doublet under the mass balance principle. We perform the numerical simulation and model discretization with the *COMSOL Multiphysics* software (COMSOL, 2021), which accounts for $P_{\text{p}}$ and associated stress and strain changes. The applied fluid-flow governing equations result from the conservation of momentum and mass in a fully saturated porous medium, whereas the heat transport governing equations are derived from the heat balance that results from both advective and conductive heat transport. We refer the reader to further details on the theory behind how fluid circu-

165 lation affects effective stresses on faults within a framework of coupled THM simulations in COMSOL (2021). Effects such as fault permeability enhancement due to the dilation, change of rock properties due to $P_{\text{p}}$ or temperature, $T$, the influence of fluid chemistry on rock mass and fault properties, mechanisms of earthquake interactions, and the Kaiser effect are not considered in the simulation.

We assume fault reactivation based on the Coulomb friction law and the notion [3] at shear slip is controlled by the ratio

170 of shear stress, $\tau$, to effective normal stresses, $\sigma_{\text{n}}'$, acting on a fault plane in the prevailing stress field, and coupled with thermo-poro-elastic response of the porous medium to the fluid circulation. The assumptions imply that fault reactivation will be initiated once $T_{\text{s}}$ exceeds the frictional resistance, $\mu$, of a fault. The size of the reactivated fault area will be restricted to the size of the critically stressed fault patch.

Within each case scenario [4]e selected two locations for our case scenarios [5]ncluding i) a fault with strike of 140°, north-

175 eastward dip angle of 60°, and an initial [6]$_{\text{s}}$ of 0.57 (referred to as 'NW-SE-striking fault'), and ii) a fault with strike of 50°, southeastward dip angle of 40°, and an initial $T_{\text{s}}$ of 0.18 (referred to as 'NE-SW-striking fault'). The simplified reservoir model has a dimensions of $20\,\text{km}$ by $20\,\text{km}$ by $8.7\,\text{km}$ and includes three stratigraphic units: impermeable overburden, representing Carboniferous layers; reservoir, representing carbonate *Massenkalk* formations of improved matrix permeability; and an impermeable underburden representing mid to early Devonian layers (Figure 3a). A fault bisects the centre of the model volume

180 between two vertical open-hole sections of production and injection wells, straddling the fault at similar distance. The wells

**Number: 1**      Author: 1 Subject: Comment on Text      Date: 15/10/2023 18:13:32
define this

> Author: kruszewski Subject: Sticky Note      Date: 04/05/2024 23:42:14
> we accept the suggestion and make appropriate changes in the manuscript.

**Number: 2**      Author: 1 Subject: Comment on Text      Date: 16/10/2023 19:11:18
you still have not defined this

> Author: kruszewski Subject: Sticky Note      Date: 04/05/2024 23:42:19
> we accept the suggestion and make appropriate changes in the manuscript.

**Number: 3**      Author: 1 Subject: Inserted Text      Date: 16/10/2023 19:23:42
suggest "theory"

> Author: kruszewski Subject: Sticky Note      Date: 04/05/2024 23:42:33
> we remove this sentence from the manuscript.

**Number: 4**      Author: 1 Subject: Inserted Text      Date: 16/10/2023 19:24:40
...the scenario of each case,

> Author: kruszewski Subject: Sticky Note      Date: 04/05/2024 23:42:42
> we accept the suggestion and make appropriate changes in the manuscript.

**Number: 5**      Author: 1 Subject: Inserted Text      Date: 16/10/2023 19:25:25
geometric configurations

**Number: 6**      Author: 1 Subject: Comment on Text      Date: 17/10/2023 15:36:11
where do the Ts of 0.57 and 0.18 come from?

> Author: kruszewski Subject: Sticky Note      Date: 04/05/2024 23:43:25
> These are the static (initial) values recomputed based on the assumed stress state and fault geometry.

[Figure]

© ⓘ
BY

are separated by 1 km and have a final depth of 4.3 km and a length of an open-hole section of 0.2 km (Figure 3b). The fault has a length of 7 km and width of 9 km and is assumed to extend from approximately 0.2 km below the surface.

We explored two different models to assess the significance of thermally induced stresses during long-term geothermal production. The first model does not account for the thermal influence resulting from an injection of a cold fluid into a hotter reservoir (isothermal model). The second model accounts for the thermal influence, where fluid flows in the subsurface with non-constant temperatures (nonisothermal model).

We assume $\mu$ of 0.6 and a cohesionless fault (Byerlee, 1978) and a geothermal gradient of $35\,°C\,km^{-1}$ (Wedewardt, 1995), with the surface temperature of $10\,°C$ and injection temperature of $50\,°C$. We model fluid circulation within a geothermal system starting in the first year of operation with duration of 25 years. We model the spatio-temporal evolution of stress for a duration of 50 years, with a time between production cessation and 50 years being considered as a period in which the reservoir returns to equilibrium conditions. Injection fluid is assumed to be clean water with its properties being temperature-dependent. Table 2 lists all other material properties assigned to the specific geological layers and the fault.

We initiate the *in situ* stress state within the model using the relations between $S_v$ and $S_{hmin}$, as well as between $S_v$ and $S_{Hmax}$ based on the mode values from Table 1. The magnitudes of the initial stress tensor are, therefore, represented by linear gradients and are artificially distributed across the model volume. The initial stress tensor constitute, therefore, merely a model input parameter. We assume the $S_{Hmax}$ azimuth in the numerical model to be equal to the [1]rcular mean value from Table 1. The bottom and side model boundaries are free to move in-plane and are fixed in the out-of-plane direction, whereas the surface is free of constraints. We impose a hydraulic head on all sides of the model and a fixed atmospheric pressure of 0.1 MPa at the surface. The open hole sections of both wells are 1D line sources, where fluid injection and production are defined with mass-flow rates and where the injection well serves as a line heat source. The model sides are open boundaries for heat transfer, whereas we fix a constant temperature at the surface and bottom boundaries, with values of $10\,°C$ and $316\,°C$ respectively.

**4 Results**

**4.1 Contemporary [2]istance to failure**

This section presents results of $T_s$, $T_d$, $S_f$, as well as the relationship between $T_s$ and $T_d$ and respective failure modes from the analytical-probabilistic model.

**4.1.1 Slip tendency**

Figure 4a and Figure 4b show the results of $T_s$ calculations as *CDF* plots. Each line in Figure 4a and Figure 4b represents one fault segment of a certain length corresponding to a fault segment presented on the map view (Figure 4c). In the *CDF* plots, more stable fault segments, i.e., ones with lower $T_s$, skew towards the left, whereas less stable fault segments, i.e., ones with higher $T_s$, skew to the right. The pink shaded areas in Figure 4a and Figure 4b show the expected range of $\mu$, being between 0.6 and 0.85 (Byerlee, 1978). The red fault segments have a conditional probability of at least 33% of their $T_s$ above the threshold

Number: 1          Author: 1 Subject: Comment on Text      Date: 17/10/2023 20:01:03
what is this, and why use this word?

Author: kruszewski  Subject: Sticky Note          Date: 04/05/2024 23:44:19
circular mean or angular mean is a mean designed for angles and similar cyclic quantities. We use the mean circular value of SHmax azimuth in our study

Number: 2          Author: 1 Subject: Highlight    Date: 04/05/2024 23:45:00
We agree with the reviewer and make appropriate changes

value of 0.6 (where $\mu$ is marked with a thick vertical black line in Figure 4a and Figure 4b), the amber fault segments have probability between 1% and 33%, and blue fault segment have less than 1% probability of being unstable.

As fault segments considered in the analysis have variable length, we refer to their combined length for further analysis. From the total combined fault length, 53% have 33% or higher probability of exceeding threshold friction, 34% have a between >1% and <33% probability, whereas 13% have a less than 1% probability. For the case of the combined length of the NW-SE-striking faults exclusively, 57% have >33% probability of exceeding threshold friction, 36% have between >1% and <33% probability, whereas 7% have <1% probability of being unstable. In the case of the combined length of NE-SW-striking faults, none have >33% probability of exceeding threshold friction, only 1% have between >1% and <33% probability, and 99% have <1% probability of being unstable.

**4.1.2 Dilation tendency**

Figures 5a and 5b show the $T_d$ calculation results as *CDF* plots and Figure 5c in map view. The *CDF* plots show fault segments with lower $T_d$ that are less prone to dilate and transmit fluids, skewed towards the left, whereas fault segments that are more likely to dilate and assist fluid flow with higher $T_d$, skewed to the right. From the total combined fault length, 63% have a probability of 33% or higher of exceeding the $T_d$ threshold of 0.8 (marked with a thick vertical black line in Figures 5a and 5b), 5% have a a probability between >1% and <33%, and 32% have a probability of less than 1%. For the case of the combined length of only NW-SE-striking faults, 67% have a probability of >33% of exceeding $T_d$ of 0.8, 5% have probability between >1% and <33%, and 28% have a probability <1%. In the case of the combined length of NE-SW-striking faults, 100% have a probability <1% of exceeding $T_d$ of 0.8.

**4.1.3 Fracture susceptibility**

Appendix A1a and A1b show the results from the $S_f$ calculations as *CDF* plots and Appendix A1c show the results in map view. In the *CDF* plots, reactivation-prone fault segments that need negligible additional pressure to become active, or have negative $S_f$ or values equal to zero, skew towards the left. Stable fault segments requiring high pressures for reactivation with positive $S_f$ values skew to the right.

From the total combined fault length, 29% have a probability of 33% or higher of $S_f$ being negative or equal to zero, 53% have a probability between >1% and <33%, and 18% have a probability of less than 1%. For the case of the combined length of the NW-SE-striking faults exclusively, 31% have probability >33% of $S_f$ being negative or equal to zero, 57% have probability between >1% and <33%, and 12% have probability <1%. In the case of the combined length of NE-SW-striking faults, no fault has probability >33% of $S_f$ being negative or equal to zero, 3% have probability between >1% and <33%, whereas 97% have <1% probability.

Number: 1     Author: 1 Subject: Comment on Text     Date: 17/10/2023 20:32:13
this does not seem to be the correct word choice here as equation is for faulting.

Author: kruszewski  Subject: Sticky Note          Date: 04/05/2024 23:45:50
This name is not given by us. It was introduced by Healy and Hicks, 2022 and we use it throughout this study.

[revised manuscript text omitted]

**Number: 1**      Author: 1 Subject: Comment on Text     Date: 17/10/2023 20:56:32

I am not proficient in the area science of thermal stress change as applied to geothermal systems but I assume that statement/conclusion if far from novel.

> Author: kruszewski   Subject: Sticky Note       Date: 04/05/2024 23:46:47
> We agree with the reviewer and make appropriate changes

**Number: 2**      Author: 1 Subject: Comment on Text     Date: 17/10/2023 21:01:37

this statement is not appropriate as you don't really address seismogenic fault rupture. I suggest to limit the verbiage to fault reactivation and rupture and to state that it is not your aim to address seismicity.

> Author: kruszewski   Subject: Sticky Note       Date: 04/05/2024 23:47:07
> We agree with the reviewer and make appropriate changes

**Number: 3**      Author: 1 Subject: Comment on Text     Date: 17/10/2023 21:00:01

what is your definition of reasonable Ts?

> Author: kruszewski   Subject: Sticky Note       Date: 04/05/2024 23:47:37
> we remove this sentence from the manuscript. Reasonable Ts is one below the frictional threshold of 0.6.

[revised manuscript text omitted]

Number: 1          Author: 1 Subject: Inserted Text          Date: 17/10/2023 21:43:54
network

> Author: kruszewski  Subject: Sticky Note          Date: 04/05/2024 23:48:02
> We agree with the reviewer and make appropriate changes

Number: 2          Author: 1 Subject: Inserted Text          Date: 17/10/2023 21:45:27
This assumption

> Author: kruszewski  Subject: Sticky Note          Date: 04/05/2024 23:49:09
> ?

Number: 3          Author: 1 Subject: Comment on Text          Date: 17/10/2023 21:51:56
there is much repetition of these conclusions throughout.  Please work to reduce this.

> Author: kruszewski  Subject: Sticky Note          Date: 04/05/2024 23:49:18
> We agree with the reviewer and make appropriate changes

are almost exclusively in
ompacting failure modes. Based on the results from the numerical modelling, we show that although

435 the contribution of the pore pressure to the fault stability is significant, it is the thermal stress resulting from rock contraction effects of the colder injection fluid in the hotter rock mass that primarily controls the size of the reactivated fault area during long-term geothermal production.

*Code availability.* The code used in this study is available at https://github.com/DaveHealy-github/pfsRuhrBochum, last access: 1st of August 2023.

440 *Data availability.* The shapefiles of faults with computed slip tendency, dilation tendency, fracture susceptibility, and reduced-risk dilation tendency based on this study, as well as the *COMSOL Multiphysics* thermo-hydro-mechanical models used to compute the spatio-temporal evolution of stress conditions *in situ* are available in Kruszewski and Verdecchia (2023).

**Appendix A: Slip, dilation tendencies and fracture susceptibility**

**A1 Slip tendency**

445 Reactivation of a fault depends on its geometry and acting stress state conditions (Barton et al., 1995). Once geometry and stress state are known, shear, $\tau$, and effective normal, $\sigma_n$', stresses exerted on any arbitrarily oriented fault can be resolved (Jaeger et al., 2007). Subsequently, in accordance with the Amonton's law, slip tendency, $T_s$, defined as a ratio between $\tau$ and $\sigma_n$', can be assessed (Morris et al., 1996)

$$T_s = \frac{\tau}{\sigma_n - P_p} = \frac{\tau}{\sigma_n'} \; . \tag{A1}$$

450 where, $\sigma_n$ is the normal stress exerted on a fault plane and $P_p$ is the pore pressure. Cohesionless (i.e., $C_0 = 0$) faults with $T_s$ approaching or exceeding frictional coefficient of sliding friction, $\mu$, have increased likelihood for a shear movement and could be deemed "unstable". The $T_s$ embodies the principle underlying Mohr-Coulomb shear failure, i.e., increasing $\tau$ relative to $\sigma_n$' causes fault to approach and eventually slip. It is assumed that fault reactivation will be initiated once $T_s$ will exceed $\mu$, which in this study is assumed to be equal to 0.6 (Byerlee, 1978)

455 $T_s \geq 0.6 \; . \tag{A2}$

**A2 Dilation tendency**

To investigate the tendency of a fault to dilate (i.e., open) under the *in situ* stress state, dilation tendency, $T_d$, is computed (Ferrill et al., 1999)

**T** Number: 1    Author: 1 Subject: Comment on Text  Date: 17/10/2023 21:56:06

thinking about this here and elsewhere in the aper, the stress conditions you specify are not sufficient to actually cause compactive failure.  I suggest that you refrain from using this or qualify it as descriptive but not physical in your geomechanical system.

Author: kruszewski  Subject: Sticky Note    Date: 04/05/2024 23:51:41

We use methodology presented in Ferrill et al., 2020 to quantify the failure modes of the mapped faults in the region. Please refer to this publication (in bibliography of our manuscript) for more details.

**T** Number: 2    Author: 1 Subject: Comment on Text  Date: 17/10/2023 21:58:34

This is strongly repetitive to the contents of the main test.  Please chose just 1 place for it.

Author: kruszewski  Subject: Sticky Note    Date: 04/05/2024 23:51:49

We agree with the reviewer and make appropriate changes

[revised manuscript text omitted]

Number: 1          Author: 1 Subject: Highlight   Date: 04/05/2024 23:52:25
We agree with the reviewer and make appropriate changes

[Figure]

[Figure]

[Figure]

**Figure 3.** Model geometry with a NW-SE-striking fault: a) discretization of the model geometry with three geological layers i.e., overburden (OB), reservoir (RES), and underburden (UB) and a fault; injection and production wells are marked in red (injection well is hidden behind the fault). The numerical model volume was discretised into approximately 730.000 elements; b) cross-section at depth of 4.2 km (i.e., reservoir depth) with the geometry of the fault zone and injection and production wells marked with red points. Fault strike is 140°, dip angle is 60°, and initial slip tendency, $T_s$, is 0.56.

[Figure]

[Figure]

[Figure]

**Figure 4.** Results of the slip tendency, $T_s$, analysis of faults in the greater Ruhr region based on the analytical-probabilistic model: a) cumulative density function (*CDF*) plot for the set of NW-SE-striking faults ($n$ = 1811). Red colour represents fault segments with >33 % chance of exceeding threshold friction, $\mu$, of 0.6 (solid black vertical line), amber shows between >1% and <33% chance, whereas blue shows <1% chance. The range of possible $\mu$ (i.e., between 0.6 and 0.85) is shown with pink shading; b) *CDF* plot for the set of NE-SW-striking faults ($n$ = 536); c) a map view of all major faults in the greater Ruhr region coloured in accordance to probabilities from a) and b). The Digital Elevation Model is from USGS (2018).

[Figure]

[Figure]

[Figure]

**Figure 5.** Results of dilation tendency, $T_d$, analysis for major faults in the greater Ruhr region based on the analytical-probabilistic model: a) cumulative density function (*CDF*) plot for the set of NW-SE-striking faults ($n = 1811$). Red colour represents fault segments with >33 % chance of exceeding threshold $T_d$ of 0.8 (solid black vertical line), amber shows between >1% and <33% chance, whereas blue shows <1% chance; b) *CDF* plot for the set of NE-SW-striking faults ($n = 536$); c) a map view of all major faults in the greater Ruhr region coloured in accordance to probabilities from a) and b). The Digital Elevation Model is from USGS (2018).

[Figure]

[Figure]

[Figure]

**Figure 6.** Dilation tendency, $T_d$, in a function of slip, $T_s$, tendency for all fault segments used in this study ($n = 2347$) with failure/reactivation modes (dashed solid horizontal black lines) and different marker colors representing a) fault dip angles and b) fault strike values. Red and blue bars represent histograms of the NW-SE- and NE-SW-striking fault segments, respectively. Dashed solid vertical black line represents the threshold sliding friction coefficient, $\mu$ of 0.6, whereas the thick solid black line, indicates $T_d$ threshold of 0.8.

[Figure]

[Figure]

**Figure 7.** a) Evolution of temperature, $T$, and b) pore pressure, $P_\mathrm{p}$, along the shortest distance, $L$, between injection and production well at a depth of $4.2\,\mathrm{km}$ for a nonisothermal coupled numerical model with NW-SE-striking fault at different simulation times; c) cumulative reactivated fault area, $A_\mathrm{r}$, (i.e., fault segments with $T_\mathrm{s} \geq 0.6$) fault for isothermal (black) and nonistohermal (red) numerical models with the injection and production flow rates, $q$, represented with blue line.

**Table 1.** Distribution of input parameters used for the analytical-probabilistic model at a reservoir depth of $4.5\,\mathrm{km}$ ($\bar{x}$ - mean value; $SD$ - standard deviation; $\kappa$ - measure of concentration; $Kurt$ - kurtosis; $P_\mathrm{p}$ - pore pressure; $S_\mathrm{hmin}$ - minimum horizontal stress magnitude, $S_\mathrm{Hmax}$ - maximum horizontal stress magnitude, $S_\mathrm{v}$ - vertical stress magnitude, $\mu$ - coefficient of sliding friction, $C_0$ - cohesion;). *Hydrostatic pressure with fluid density of $1000\,\mathrm{kg\,m^{-3}}$; **Extracted from geological maps.

| Property | $\bar{x}$ | $SD$, $\kappa$ | Unit | Distribution | Reference |
|---|---|---|---|---|---|
| $P_\mathrm{p}$* | 44 | 2.2 (5% of mean) | MPa | Gaussian | - |
| $S_\mathrm{hmin}$ | 65 | 6.5 (10% of mean) | MPa | Skewed normal ($Kurt$ = +3.5) | Kruszewski et al. (2022a) |
| $S_\mathrm{Hmax}$ | 114 | 26 (22.5% of mean) | MPa | Skewed normal ($Kurt$ = +2.9) | Kruszewski et al. (2022a) |
| $S_\mathrm{v}$ | 111 | 6 (5% of mean) | MPa | Gaussian | Kruszewski et al. (2022a) |
| Azimuth of $S_\mathrm{Hmax}$ | 161 | 30 | ° | Von Mises (circular normal) | Kruszewski et al. (2022a); Heidbach et al. (2018) |
| Strike** | - | - | ° | - | GD NRW (2017, 2019) |
| Dip (NW-SE) | 65 - 85 | - | ° | Uniform | Jansen et al. (1986); Drozdzewski et al. (2007) |
| Dip (NE-SE) | 35 - 60 | - | ° | Uniform | Jansen et al. (1986); Drozdzewski et al. (2007) |
| $\mu$ | 0.6 | 0.1 (10% of mean) | - | Gaussian | Byerlee (1978) |
| $C_0$ | 1 | 0.2 (20% of mean) | MPa | Gaussian | Wyllie and Norrish (1996) |

[Figure]

[Figure]

[Figure]

**Figure 8.** Spatial changes of shear stress, $\tau$, effective normal stress, $\sigma_n$', and slip tendency, $T_s$, on a NW-SE-striking fault with strike of $140°$, dip angle of $70°$, and an initial $T_s$ of 0.57 in the greater Ruhr region presented for the case of 5th (a, d, g), 25th (b, e, h), and 50th (c, f, i) simulation year (*OB* - overburden; *RES* - reservoir; *UB* - underburden); Area of the fault with $T_s \geq 0.6$ is marked with a black outline in g), h), and i).

[Figure]

[Figure]

[Figure]

**Figure 9.** Results of the reduced-risk dilation tendency, $T_{dn}$, analysis for all major faults in the greater Ruhr region ($n = 2347$). The stereonet of $T_{dn}$ for any arbitrarily oriented fault in the greater Ruhr region is located in the top left corner of the map. The Digital Elevation Model is from USGS (2018).

[Figure]

[Figure]

**Table 2.** Parameters used for the coupled thermo-hydro-mechanical numerical simulations, where $d$ - thickness; $p_B$ - bulk rock density; $E_s$ - static Young's modulus; $\nu$ - Poisson's ratio; $\beta$ - Biot coefficient; $k$ - matrix permeability; $\phi$ - porosity; $\lambda$ - thermal conductivity; $\alpha$ - coefficient of thermal expansion; $C_p$ - heat capacity at constant pressure; [a]Drozdzewski et al. (2007), [b]Jansen et al. (1986), [c]DEKORP (1990), [d]Farkas et al. (2021), [e]RUBITEC (2003), [f]Brenne (2016), [g]Lippert et al. (2022), [h]Manger et al. (1963), [i]Balcewicz et al. (2021), [j]Mullen et al. (2007), [k]Robertson (1988), [l]Griffith (1936).

| Property | Overburden | Reservoir | Underburden | Fault |
|---|---|---|---|---|
| $d$ (m) | [a, b, c]4000 | [a, b, c]700 | [a, b, c]4000 | [d]1E-4 |
| $\rho_B$ (kg m$^{-3}$) | [e, f]2500 | [g]2604 | [h]2504 | 2000 |
| $E_s$ (MPa) | [e]52.5 | [i, j]50.0 | [f]29.0 | - |
| $\nu$ (-) | [e]0.25 | [i]0.33 | [f]0.17 | - |
| $\beta$ (-) | 1 | 1 | 1 | - |
| $k$ (m$^2$) | [e]1E-18 | 1E-14 | [e]1E-18 | 1E-11 |
| $\phi$ (-) | [e]0.01 | [g]0.04 | [f]0.02 | 0.50 |
| $\lambda$ (W m$^{-1}$K$^{-1}$) | [e]3.6 | [g]3.1 | [f]3.5 | 3.0 |
| $C_p$ (J kg$^{-1}$K$^{-1}$) | [e]818 | [k]880 | [k]880 | 800 |
| $\alpha$ (K$^{-1}$) | [l]1.5E-5 | [k]8E-6 | [l]1.5E-5 | - |

[Figure]

[Figure]

[Figure]

**Figure A1.** The results of fracture susceptibility, $S_f$, for major faults in the greater Ruhr region based on the analytical-probabilistic model: a) cumulative density function (*CDF*) plot for the set of NW-SE-striking faults ($n = 1811$). Red colour represents fault segments with >33 % chance of exceeding threshold $S_f$ of 0 MPa (solid black vertical line), amber shows between >1% and <33% chance, whereas blue shows <1% chance; b) *CDF* plot for the set of NE-SW-striking faults ($n = 536$); c) a map view of all major faults in the greater Ruhr region coloured in accordance to probabilities from a) and b). The Digital Elevation Model was sourced from USGS (2018).

[Figure]

[Figure]

**Figure A2.** Spatial changes of shear stress, $\tau$, effective normal stress, $\sigma_n'$, and slip tendency, $T_s$, on a NE-SW-striking fault with strike of 70° and dip angle of 50° and an initial $T_s$ of 0.18 in the greater Ruhr region presented for the case of 5th (a, d, g), 25th (b, e, h), and 50th (c, f, i) simulation year (*OB* - overburden; *RES* - reservoir; *UB* - underburden).

---

## Author Comment (AC3)

Dear Reviewer, we are very grateful for the careful revision of our manuscript and for making our study more complete. We benefited greatly from your feedback and, after careful revision and implementation of your comments and critiques, we are coming back with the manuscript revision and detailed replies to your comments. Please let us know in case of any questions or concerns regarding the new version of the manuscript or the replies below. Our replies are in red and your comments are in black.

This manuscript presents approaches and results very similar to previous studies without relevant increase of information. The concepts and ideas are already well established. Along the manuscript, necessary details and quality controls are missing, especially regarding the numerical reservoir modelling, which prevents from being confident in the quantified results. Moreover, the deterministic THM modelling remains very generic and therefore delivers common knowledge and general results. The scientific significance is missing over the manuscript. Besides, the manuscript is wordy and extrapolates the results. Consequently, I recommend that this manuscript be rejected for publication in Solid Earth and any other journal.

The methodology and the results presented have been discussed in a very similar way and for the same area by the same main author in:

- Kruszewski, M., G. Montegrossi, T. Backers, and E. H. Saenger, 2021, In Situ Stress State of the Ruhr Region (Germany) and Its Implications for Permeability Anisotropy: Rock Mechanics and Rock Engineering, **54**, 6649–6663.

- Kruszewski, M., G. Klee, T. Niederhuber, and O. Heidbach, 2022a, In situ stress database of the greater Ruhr region (Germany) derived from hydrofracturing tests and borehole logs: Earth System Science Data, **14**, 5367–5385.

- Kruszewski, M., G. Montegrossi, M. Balcewicz, G. de Los Angeles Gonzalez de Lucio, O. A. Igbokwe, T. Backers, and E. H. Saenger, 2022b, 3D in situ stress state modelling and fault reactivation risk exemplified in the Ruhr region (Germany): Geomechanics for Energy and the Environment, **32**, 100386.

Despite the probabilistic approach was applied in this manuscript, the added-value is questionable.

Our study investigates in greater detail the probability for fault reactivation utilizing new probabilistic approaches based on known geological uncertainties from the region including the recent comprehensive stress database from the Ruhr region with more than 420 stress magnitude measurements and detailed fault maps created from an abundance of geological data from the coal mining. Such number of geological parameters as well as the new approach to the probabilistic analysis with three scalar values and their thresholds makes our analysis unique. Additionally, to the probabilistic analysis of slip tendency, dilation tendency, and fracture susceptibility, we use the slip vs dilation parameter space to quantify the failure modes of different faults in the region. On top of that, we introduce new scalar value for an improved prospecting of structurally-controlled geothermal resources, called reduced-risk dilation tendency, that allows for quick scanning for structures less likely to slip in shear and more likely to dilate and serve as fluid conduits. Additionally, we use numerical modelling approaches to quantify the evolution of slip and dilation tendency of a fault during long-term geothermal production, quantifying also the change of the failure mode of the fault patch. These components are new, innovative, and have not been yet tackled in the scientific literature as well as were they not part of any of the abovementioned studies mentioned by the reviewer. We consider all the mentioned components of our study to present valuable contribution into the field of reservoir geomechanics and geothermal prospecting in greenfield areas.

2. Regarding the probabilistic approach, why the procedure presented by Seithel *et al.* (2019) (a paper your refer to) is not used although it was developed for the same purpose?

We use (and also update) the methodology and open access *Python* code developed by *Healy and Hicks (2022)* and not the methodology developed in *Seithel et al. (2019)*. The main advantage of the methodology developed in *Healy and Hicks (2022)* is that it is based on a combined Monte Carlo, response surface methodology, and Mohr-Coulomb theory, where the three scalar values (Ts, Td, Sf) for the fault stability analysis are computed for a set of multiple fault segments simultaneously accounting for known geological uncertainties within the reservoir. With their methodology a large amount of simulations can be carried out for a single fault. The results from Python code by *Healy and Hicks (2022)* can be easily transported into maps of probability, like presented in our manuscript. We find, therefore, approach by *Healy and Hicks (2022)* to more applicable for our study.

3. The slip tendency results (Fig. 4) highlight fault patches that have Ts higher than 1. How can it be?

The probabilistic approach considers all model uncertainties which result for extreme cases in faults that are Ts > 1. This would imply that the fault is at failure and should dissipate accumulated elastic stress by failure (either seismic or aseismic slip). However, such a process is not part of the model. The situation of Ts greater than 1 has occurred in a very few cases where normal stress exerted on the fault approached the value of the shear stresses on a fault. It is due to the very few Monte Carlo simulations having large sigma1 values but very low sigma3 values (which are few of the data outliers from the assumed data distribution presented in Figure 2 and then on CDF plots in Figure 3a and b). It should be also mentioned that for the further analysis (maps of fault stability etc.), we take into consideration the probability of Ts exceeding 0.6 and not the few data outliers approaching 1.0. In reality, during e.g., geothermal production Ts will approach or even exceed 1.0 due to the significant lowering of the normal stresses resulting from poro-thermo-elastic effects (as presented in Figure 9a). In short, these are few data outliers resulting from the assumed distributions of principal stresses that have no strong impact on the simulation results.

4. With the presented results, many faults should be already critically stressed. How do you explain that no natural seismicity is observed in the area? A chapter discussing the natural seismicity of the area is missing.

In the introduction as well as in the discussion part of the manuscript, we briefly discuss the issue of seismicity in the Ruhr region (L51-53). The seismicity is primarily anthropogenic and connected to either quarry blasting or coal mining (and the recent mine flooding activities). The greater Ruhr region can be considered, therefore, as an aseismic or seismically quiescent region. In the discussion part, we discuss the recent seismic events related to the mine flooding activities in the old coal mines, where relatively small amount of pressures (1 MPa) allowed to create large seismic events of ML 2.6, which we believe to be related to the major fault structures (L193-198). This could indicate that the faults (NW-SE striking) in the region are either critically stressed or close to being critically stressed. The lack of seismicity in the region can be, however, explained by e.g., a release of seismic energy with aseismic fault movement i.e., fault creep. More studies are, however, need to prove this theory.

5. For the range of pressure found in the Ruhr area at that depth (<2 kbar), Byerlee (1978) observed friction coefficients of 0.85. However, the limit assumed in the manuscript is 0.6 (L132, L187, Table 2) and not 0.85, using the same reference, why? L210, however, refereeing to Byerlee (1978) again, the 0.85 friction is written to be a possible value!

The nearly 50-year-old *Byerlee (1978)* paper has included static friction coefficient of many different rock types carried out using different types of laboratory testing techniques. The function of static friction coefficient introduced by Byerlee is rather a rough approximation of rock friction independent of rock type or temperature and pressure conditions. We decided to use the static coefficient of 0.6, as suggested by e.g., *Zoback (2009)* and many other scholars in reservoir geomechanics. The assumption of static friction coefficient of 0.6 is, however, not at all random, but it was based on indications of static frictional properties of rocks i) in the Ruhr region and ii) carbonate rocks. We show below examples from the literature of static friction coefficients computed based on either field or laboratory data. Based on the studies below, it can be seen that the static friction coefficient will rarely exceed 0.85 and will be in range between 0.6 and 0.7. This means that the frictional properties of rock will depend significantly on the type of rock as well as on the temperature and pressure conditions expected in the reservoir. We, therefore, still believe that the assumption of static friction coefficient of 0.6 for the Devonian carbonates, made in our study, is appropriate. Laboratory tests on samples from the region, however, are needed to prove this value.

In L162-163 (in the new version of the manuscript) as well as in Figure 3 we merely show the allowable range of static friction coefficient as indicated by *Byerlee (1978)* and not the friction value assumed in this study. The assumed value of 0.6 is indicated in the text (L162) and in Table 1. We add to the Table 1 the references included below. Below we include static friction coefficient estimated from hydrofracturing tests in the Ruhr region (performed at depths of coal mines in the region until approximately 1.4 km depth) from *Kruszewski et al. (2022)*:

[Figure]

Below a figure from *Pluymakers et al. (2016)* showing static friction coefficient for different carbonate rocks.

[Figure]

[1] Morrow et al, 2000; Shimamoto and Logan, 1981; Verberne et al, 2010, 2013, 2015
[2] Scuderi et al, 2013; Shimamoto and Logan, 1981; Pluymakers et al, 2014; Pluymakers and Niemeijer, 2015; Pluymakers et al, current study.
[3] Scuderi et al, 2013; Shimamoto and Logan, 1981; Pluymakers et al, current study; Weeks and Tullis, 1985; Samuelson et al, pers.comm.
[4] Scuderi et al, 2013, Pluymakers et al, current study.

Below a figure from *Hunfeld et al. (2017)* showing static coefficient of friction for rocks in the Groningen field in the Netherlands.

[Figure]

6. For Sf, in Eq. 3, Co is accounted for, but it does not appear for the slip tendency although both parameters (Ts and Sf) have the same theoretical background (Mohr-Coulomb failure criterion). Why is it so?

We agree with the comment. We remove cohesion from the analysis and from the manuscript focusing on an idealized case of a cohesionless fault(s). The main reason from that is that the available data on cohesion from the region is, as of now, nonexistent. As a result, we amend calculations of Sf and include these changes in the manuscript. Due to this, we provide new maps of Sf in the revised version of this manuscript.

7. In section 5.2, L378-379: "*Scalar values used for fault stability evaluation based on the contribution of fluid pressure only, such as Sf, will not provide a full picture of the fault stability in situ*". This is also true for the slip tendency, so mention it as well.

We agree with the comment and make appropriate changes to the manuscript by removing the sentence.

8. The last sentence of the conclusion is not surprising and does not need any result of the numerical simulation that was described in the manuscript. Below is a (non-exhaustive) list of papers that are already

discussing the importance of thermally induced stress changes on a long term basis in geothermal contexts:

- De Simone, S., V. Vilarrasa, J. Carrera, A. Alcolea, and P. Meier, 2013, Thermal coupling may control mechanical stability of geothermal reservoirs during cold water injection: Physics and Chemistry of the Earth, Parts A/B/C, **64**, 117–126.

- Egert, R., Gaucher, E., Savvatis, A., Goblirsch, P., Kohl, T., 2022. Numerical determination of long-term alterations of THM characteristics of a Malm geothermal reservoir during continuous exploitation. Presented at the European Geothermal Congress 2022, Berlin, Germany.

- Jeanne, P., J. Rutqvist, and P. F. Dobson, 2017, Influence of injection-induced cooling on deviatoric stress and shear reactivation of preexisting fractures in Enhanced Geothermal Systems: Geothermics, **70**, 367–375.

- Jeanne, P., J. Rutqvist, P. F. Dobson, J. Garcia, M. Walters, C. Hartline, and A. Borgia, 2015, Geomechanical simulation of the stress tensor rotation caused by injection of cold water in a deep geothermal reservoir: Journal of Geophysical Research: Solid Earth, **120**, 8422–8438.

- Kivi, I. R., E. Pujades, J. Rutqvist, and V. Vilarrasa, 2022, Cooling-induced reactivation of distant faults during long-term geothermal energy production in hot sedimentary aquifers: Scientific Reports, **12**, 2065.

- Koh, J., H. Roshan, and S. S. Rahman, 2011, A numerical study on the long term thermo-poroelastic effects of cold water injection into naturally fractured geothermal reservoirs: Computers and Geotechnics, **38**, 669–682.

- Wassing, B. B. T., T. Candela, S. Osinga, E. Peters, L. Buijze, P. A. Fokker, and J. D. Van Wees, 2021, Time-dependent Seismic Footprint of Thermal Loading for Geothermal Activities in Fractured Carbonate Reservoirs: Frontiers in Earth Science, **9**.

Many references regarding THM modelling in similar contexts should be given but they are missing. They could have been used as inspiration source.

We have added few of the mentioned references to the manuscript (L67-68). We have changed the rationale of the study, where now we do not discuss the thermal effects on fault reactivation but rather focus on the temporal evolution of slip and dilation tendencies of the fault during long-term geothermal production to show the evolution of the reactivation potential in time (and space) as well as the change of the fault failure conditions on the Ts vs. Td parameter space. Thermal effects, as pin-pointed by the reviewer, is a widely known phenomenon already discussed in the literature and, therefore, we decide to not discuss it in the new version of the manuscript.

9. When presenting THM numerical modelling, it is necessary to develop much more what is actually done to give confidence in the results. So far, it is not the case, and a lot of information is missing, e.g. what are the physical processes activated (equations)? The above-mentioned papers could help to do so.

As already mentioned, we change the rationale of our study, where numerical modelling is not anymore main part of the methodology and not main part of the study. Numerical model is now included just in the discussion part of the manuscript (Section 5.4) and is used only to support the arguments stated in the main parts of the study. We decide, therefore, not to discuss in extensive detail all the physical processes and equations used in the numerical model as we deem it redundant. We describe what

processes control the developed numerical model, what are the input parameters and what are the boundary conditions used, both in the text (Section 5.4; L285-307) and in improved Figure 8. In the *Data Availability* section of the manuscript we publish (open access) the developed numerical models (see *Kruszewski and Verdecchia (2023)* in the references), including all necessary information about physical processes, equations, input parameters, boundary conditions, discretization etc., with the manuscript to allow reproducibility of the results presented in our study. The updated numerical models will be published with the revised version of this manuscript. These models can be easily checked by readers or reviewers for reproducibility. For more detailed explanations of the physical processed/equations used in the numerical model we refer the reviewer to *COMSOL (2021)* as well as *Taillefer et al. (2018)*, both are referenced in the manuscript.

10. Was a mesh sensitivity study carried out? This is questionable when looking at the discontinuous curves of Fig. 7a and b.

Yes, we have carried out convergence tests on models three different mesh sizes. Below a snapshot comparing results of the cumulative reactivated fault area (Ar; on the left) as well as the maximum dilation tendency on the fault plane (on the right) computed with the model discretized into $0.73 \cdot 10^6$ elements and one discretized in $1.28 \cdot 10^6$ elements. We skip the model with $0.3 \cdot 10^6$ elements, where model results were deemed to be dependent on mesh size. The results we deemed to be satisfactory and we use the model discretized in $1.28 \cdot 10^6$ elements for our discussion in the manuscript.

[Figure]

11. In the THM results, it would be most important to see space and time distribution of, at least, the pore-pressure field and the temperature field before jumping directly to the shortest distance between wells and fault.

We make changes to our approach and we use now numerical modelling only in the discussion part of the study to show the evolution of both slip and dilation tendencies as well as to compute the cumulative reactivated fault area and fault area with dilation tendencies larger than 0.8. We decided not to discuss the thermal and pore pressure effects at all in the manuscript. This, as pin-pointed by the reviewer, is widely known phenomenon already discussed in the literature. We decide, therefore, to not include the time and space evolution of pressure and temperature fields as advised by the reviewer.

12. Section 3.2, L165-168: "*Effects such as fault permeability enhancement due to the dilation, change of rock properties due to Pp or temperature, T, the influence of fluid chemistry on rock mass and fault properties, mechanisms of earthquake interactions, and the Kaiser effect are not considered in the simulation*" This looks like COMSOL could account for all of these. I am not aware that COMSOL can simulate earthquakes.

We do not account for the listed effects in our study and it is mainly due to the lack of published data from the region on fault permeabilities, change of rock properties with T,P conditions, fluid chemistry, Kaiser effect, earthquake interaction and so on. Performing simulation with all listed effects will lead to a significantly overconstrained model with much larger uncertainties, which was not our aim in this study. With our analysis, using simple and idealized numerical models, based on limited input and geological data from the geothermal reservoir, we investigate the possible fault reactivation and the evolution of slip and dilation tendency in time and space. Although, we would like to add all of the listed effects to our work in the future, when more data will become available from the region, as of now, we find that the simulation results presented in our study still give a good picture of what could occur in the subsurface with the amount of data that is, as of now, available.

13. First sentence of abstract: This is wrong as underlined e.g. by the deep geothermal exploitation in the Paris basin for many decades.

We agree with the comment. We make it more precise now and include a part where we say that our analysis tackles only structurally controlled geothermal systems (where the matrix permeability in insufficient for geothermal fluid production; L25-30). This has been made also now clearer with the new title of our manuscript.

14. Nothing in the manuscript supports the simulation of seismicity or aseismic slip or seismic hazard. Consequently, these aspects should be mentioned with care.

We agree with the comment and make appropriate changes to the manuscript.

15. Second sentence of abstract: what is the Earth's "plumbing" system? I have never read such wording in a geothermal context. Do you mean "circulation"?

We agree with the comment and change the phrasing in the manuscript. By *plumbing system*, we meant the *hydraulic system*, but we agree that it was rather colloquial phrasing (L1).

16. L18: […] a complex "web" of faults […]? I have never read such wording in a geothermal context. Do you mean "network"? It is found L401 as well.

We agree with the comment and change the phrasing in the manuscript. We meant fault network.

17. Avoid using "the distance to failure" (e.g. L31), you mean in meters (?), prefer the "reactivation potential".

We agree with the comment and change the phrasing in the manuscript.

18. The Appendix does not correspond at all to what is announced in the main part of the manuscript.

We agree with the comment. We integrate the appendix into the main text and amend the text accordingly (L115-136).